



# GCL-Mascon2024: a novel satellite gravimetry mascon solution using the short-arc approach

Zhengwen Yan[1], Jiangjun Ran[1], Pavel Ditmar[2], C K Shum[3], Roland Klees[2], Patrick Smith[3], and Xavier Fettweis[4]

[1]Department of Earth and Space Sciences, Southern University of Science and Technology, Shenzhen, 518055, P. R. China
[2]Department of Geoscience and Remote Sensing, Delft University of Technology, Delft, 2628 CN, The Netherlands
[3]Division of Geodetic Science, School of Earth Sciences, The Ohio State University, Columbus, Ohio 43210, USA
[4]Department of Geography, University of Liège, Liège, B-4000, Belgium

*Correspondence to*: Jiangjun Ran (ranjj@sustech.edu.cn)

**Abstract.** This paper reports an innovative mass concentration (mascon) solution obtained with the short-arc approach, named "GCL-Mascon2024", for estimating spatially enhanced mass variations on the Earth's surface by analyzing K-/Ka Band Ranging satellite-to-satellite tracking data collected by the Gravity Recovery And Climate Experiment (GRACE) mission. Compared to contemporary GRACE mascon solutions, this contribution has three notable and distinct features: First, this solution recovery process incorporates frequency-dependent data weighting techniques to reduce the influence of low-

frequency noise in observations. Second, this solution uses variable-shaped mascon geometry with physical constraints such as coastline and basin boundary geometries to more accurately capture temporal gravity signals while minimizing signal leakage. Finally, we employ a solution regularization scheme that integrates climate factors and cryospheric elevation models to alleviate the ill-posed nature of the GRACE mascon inversion problem. Our research has led to the following conclusions: (a) the temporal signals from GCL-Mascon2024 exhibit 6.5%−20.4% lower residuals over the continental regions, as compared

with the (Release) RL06 versions of other contemporary mascon solutions from GSFC, CSR, and JPL; (b) in Greenland and global hydrologic basins, the correlation coefficients of estimated mass changes between GCL-Mascon2024 and other RL06 mascon solutions exceed 95.0%, with comparable amplitudes; especially over non-humid river basins, the GCL-Mascon2024 suppresses random noise by 36.7% compared to contemporary mascon products; and (c) in desert regions, the analysis of residuals calculated after removing the climatological components from the mass variations indicates that the GCL-

Mascon2024 solution achieves noise reductions of over 28.1% as compared to the GSFC and CSR RL06 mascon solutions. The GCL-Mascon2024 gravity field solution (Yan and Ran, 2024) is available at https://doi.org/10.5281/zenodo.14008167.

## 1 Introduction

Comprehending the Earth as a dynamic system relies heavily on our knowledge of its gravity field, mass variations induced
by fluid layers, as well as geophysical or climatic processes (e.g., Wahr et al., 1998; Pail et al., 2015). Over the past two
decades, significant achievements have been made through the availability of observations collected by satellite gravimetry
missions, such as the Gravity Recovery And Climate Experiment (GRACE) (Tapley et al., 2004; Tapley et al., 2019) and its
successor, GRACE Follow-On (GRACE-FO) (Flechtner et al., 2016; Landerer et al., 2020). These satellite gravity missions
have not only enhanced our understanding of temporal variations in the Earth's gravity field but also played a crucial role in
advancing various disciplines, including glaciology, hydrology, geophysics, oceanography, atmosphere, and climate science
(e.g., Han et al., 2006a; Chen et al., 2009; Rignot et al., 2011; Jacob et al., 2012; Rodell et al., 2018).

Gravity field variations expressed in spherical harmonics have been extensively and widely employed in satellite geodesy for
decades (Chen et al., 2022). However, certain limitations persist in the application of spherical harmonic solutions from
GRACE/GRACE-FO data, including the presence of north-south "stripes" (Swenson and Wahr, 2006) as well as signal leakage
(Kusche et al., 2009), particularly in regions adjacent to the land-sea boundary. The main reasons for the above problems are
temporal aliasing (Wiese et al., 2011b) and the design of the satellite orbits and tracking systems (Wiese et al., 2011a),
including inclination, altitude, inter-satellite distance, and co-planar low-low satellite-to-satellite tracking system.
Conventional approaches involve the removal of "stripes" through empirical smoothing (e.g., Wahr et al., 1998), de-striping
(e.g., Swenson and Wahr, 2006), or regularization techniques (e.g., Save et al., 2012). It is important to note that though these
methods are largely effective in preserving signals and suppressing noise, the elimination of stripes also results in a reduction
in the genuine geophysical signals (e.g., Han et al., 2005; Yi and Sneeuw, 2022; Zhou et al., 2023). Moreover, the efficacy of
destriping is highly dependent on the characteristics of the signals, including their size, shape, and orientation (Watkins et al.,
2015). It is worth mentioning that the impact of aliasing errors can be mitigated by combining gravity satellite formations
within optimal constellation configuration (Yan et al., 2024) or by recovering the temporal gravity field at a higher temporal
resolution (Yan et al., 2023).

Alternatively, mass concentration (mascon) solutions can be utilized to model the temporal gravity field. This technique was
initially introduced by Muller and Sjogren (1968) in their efforts to develop a model for the static gravity field of the Moon.
Whereafter, mascon solutions utilizing GRACE Level-1B data were initially conducted in a regional context (e.g., Rowlands
et al., 2005; Luthcke et al., 2006) and subsequently extended to encompass diverse global parameterizations (e.g., Luthcke et
al., 2013; Watkins et al., 2015; Save et al., 2016; Allgeyer et al., 2022). Besides, some attempts have been made to enhance
mascon solutions' spatial (e.g., Loomis et al., 2021) or temporal resolution (e.g., Croteau et al., 2020). Afterward, to mitigate
the computational complexity, alternative variants of the mascon approach have been put forward, which utilize monthly sets
of spherical harmonic coefficients (SHCs, i.e., Level-2 data) as input (e.g., Forsberg and Reeh, 2006; Baur and Sneeuw, 2011;
Schrama and Wouters, 2011). Numerous recent publications have used mascon solutions released by responsible agencies,
including the NASA (National Aeronautics and Space Administration) Goddard Space Flight Center (GSFC), the NASA Jet
Propulsion Laboratory (JPL), and the University of Texas at Austin Center for Space Research (CSR) in the United States.
The mascon solution released by JPL (JPL RL06 mascon) utilizes explicit partial derivatives with analytical expressions for
the mascons to establish the relationship between inter-satellite range-rate measurements and individual mascons (Wiese et al.,
2018), whereas the latest variants of GSFC mascon solutions (GSFC RL06 mascon) and CSR mascon solutions (CSR RL06
mascon) are characterized by a finite series of spherical harmonic functions, with the corresponding partial derivatives
computed using the chain rule (Loomis et al., 2019; Save, 2020). These GRACE/GRACE-FO gravimetry data processing
centers also offer visualization tools for their mascon products, facilitating analysis and comparison of the latest mascon
solutions as well as generating time series data for specific regions.

Various methods including the dynamic approach (e.g., Kvas et al., 2019), the short-arc approach (e.g., Mayer-Gürr, 2008), the celestial mechanics approach (e.g., Beutler et al., 2010), the energy balance approach (e.g., Han et al., 2006b), and the acceleration approach (e.g., Ditmar and Van Der Sluijs, 2004), play a vital role in modeling the temporal gravity field from level-1B satellite gravimetry data. To date, most publicly available global mascon products based on Level-1B data commonly rely on longer arcs (e.g., 24-hr ones). This includes the mascon solutions recovered using the dynamic approach by GSFC (Loomis et al., 2019), CSR (Save, 2020), and JPL (Watkins et al., 2015), as well as the mascon solution by the Australian National University (ANU) utilizing the celestial mechanics approach (Allgeyer et al., 2022; Tregoning et al., 2022; Mcgirr et al., 2023). This study represents the first application of the short-arc approach to recover the global mascon solution.

Frequency-dependent noise in GRACE measurements significantly limits GRACE from reaching the prelaunch baseline accuracy; thus, modeling this noise is a critical aspect for improving the accuracy of temporal gravity field recovery. In the context of spherical harmonic coefficient solutions, the impact of frequency-dependent noise in observations is typically accounted for by introducing empirical parameters (Liu et al., 2010; Zhao et al., 2011) to absorb errors or by using frequency-dependent data weighting (FDDW) techniques (Klees et al., 2003; Ditmar et al., 2007). However, the potential of suppressing frequency-dependent errors in mascon modeling with the FDDW technique remains largely unexplored.

Herein, the Geodesy and Cryosphere Laboratory (GCL) from the Southern University of Science and Technology has released a new series of mascon solutions (hereafter referred to as GCL-Mascon2024) using the short-arc approach and FDDW, as well as advanced regularization schemes. These mascon solutions incorporate pertinent physical constraints to estimate global mass variations directly from inter-satellite range-rate measurements. To alleviate the effects of errors introduced by signal leakage, the GCL-Mascon2024 solution employs a strategy that involves segmenting the mascon shape based on land-sea boundaries and the boundaries of distinct hydrologic basins. Subsequently, this paper aims to investigate the impact of selecting arc length and accelerometer calibration parameters in the short-arc approach on the mascon solutions while also providing a quantitative evaluation of the GCL-Mascon2024 solution.

The article is organized as follows. Section 2 describes the methodology for recovering global mascon solutions with the short-arc approach. Section 3 discusses the parameter determination on global mascon solutions using the short-arc approach. Section 4 evaluates the scientific results of real data processing with the proposed approach. Section 5 provides detailed information and links for accessing the dataset utilized in this study, along with the GCL-Mascon2024 solution released in this work. Finally, section 6 provides the main conclusions.

## 2 Methodology

Building upon the earlier studies by Ran et al. (2018) and Ran et al. (2021), we propose a new mascon approach recovered from GRACE Level-1B tracking data based on the short-arc approach. The primary distinction between GCL-Mascon2024 and the aforementioned mascon solutions lies in the type of exploited input data (i.e., Level-1B vs. Level-2). The mascon solutions released by Ran et al. (2018) and Ran et al. (2021) are based on spherical harmonic coefficients and cover only mass anomalies over Greenland. The GCL-Mascon2024 solution is a series of global mascons with analytical partial derivatives. In other words, we establish a direct relationship between the mass variations of mascons and the inter-satellite measurements. Section 2.1 elaborates on the utilized functional model, which links GRACE Level-1B data to mascon solutions. Section 2.2 outlines the strategy for defining mascon geometry during the data inversion process. Section 2.3 describes the background force models and input data employed to recover GCL-Mascon2024 solution. Section 2.4 explains suppressing frequency-dependent errors by using the FDDW technique. Finally, the advanced spatial constraints exploited in the inversion procedure are presented in section 2.5.





## 2.1 Mathematical Formulation

A satellite in orbit around the Earth is subject to gravitational forces which are governed by Newton's law of universal
gravitation. The temporal gravity field can be modeled as a series of $N$ mascons, with the surface mass density (mass per unit
area) of mascon $M_i$ represented by $\rho_i$ ($i$=1, 2, ..., N). When the satellite is at measurement point $p$, the gravitational forces $\boldsymbol{f}_p$
exerted on the satellite by the mass variations of the Earth's surface can be expressed as

$$\boldsymbol{f}_p = G \sum_{i=1}^{N} \rho_i \int_{M_i} \frac{\hat{\boldsymbol{d}}_p \cdot ds}{\left(l_p\right)^2} = G \sum_{i=1}^{N} \rho_i \cdot \hat{\boldsymbol{I}}_{i,p}. \tag{1}$$

Here, G is the universal gravitational constant; $\hat{\boldsymbol{d}}_p$ is the unit vector directed from the satellite measurement point toward the
surface mass; Define $l_p$ the distance between the satellite measurement point $p$ and an integration point on mascon; $\hat{\boldsymbol{I}}_{i,p}$ is a
vector pointing from the satellite measurement point $p$ to the given mascon $M_i$, which is calculated using numerical integration.
To that end, we utilize a composed Newton-Cotes formula (Gonzalez, 2010) applied to the Fibonacci nodes, i.e., the Fibonacci
nodes as integration points mentioned aforementioned. By defining the surface area and the number of the Fibonacci nodes of
mascon $M_i$ as $S_i$ and $K_i$, we can calculate $\hat{\boldsymbol{I}}_{i,p}$ as

$$\hat{\boldsymbol{I}}_{i,p} \approx \sum_{j=1}^{K_i} \frac{S_i}{K_i \cdot \left(l_{ij,p}\right)^2} \cdot \hat{\boldsymbol{d}}_{ij,p}, \tag{2}$$

where $l_{ij,p}$ represents the distance between a Fibonacci point $j$ located in the mascon $M_i$ and the satellite measurement point $p$;
$\hat{\boldsymbol{d}}_{ij,p}$ is a unit vector pointing from the satellite measurement point $p$ to a Fibonacci point $j$ located in the mascon $M_i$.
Then,

$$\boldsymbol{f}_p = \sum_{i=1}^{N} \underbrace{\rho_i}_{\boldsymbol{x}} \cdot \underbrace{G \frac{S_i}{K_i} \cdot \sum_{j=1}^{K_i} \frac{\hat{\boldsymbol{d}}_{ij,p}}{\left(l_{ij,p}\right)^2}}_{\boldsymbol{G}_p}. \tag{3}$$

Combining $\boldsymbol{G}_p$ over multiple positions/epochs within an arc yields the matrix $\boldsymbol{G}$ which is used in the observation model
(Mayer-Gürr, 2008) with orbit and range-rate measurements as observation types.

## 2.2 Parameterization

The choice of an appropriate mascon partitioning strategy is crucial for mitigating noise amplification during the data inversion
process (Ran et al., 2018). In this study, the selection of mascon geometry is based on incorporating pertinent physical
constraints, such as the geometry of the coastal line and basin boundaries. The definitions of these basin boundaries are derived
from Scanlon et al. (2018). Regarding the aforementioned parameterization, the primary assumption is that there is no signal
correlation between mascons located in different basin systems (Ran et al., 2021), meaning that basins do not share mascons
with their neighboring basins to reduce signal leakage between the corresponding basins.

In the GCL-Mascon2024 processing scenario, the estimated monthly mascon solution has a spatial resolution of about 300×300
km and 400×400 km on land and ocean, respectively. The total number of mascons is 4069, with 1879 terrestrial mascons and
2217 ocean mascons. Figure 1 provides the mascon partitioning of GCL-Mascon2024. It is important to note that the mascons
located within the basins and coastal regions are defined in accordance with the boundary geometry. The numerical integration
points, as discussed in section 2.1, are distributed on a Fibonacci grid with an average spacing of 10 km, requiring the
generation of approximately 5.1 million Fibonacci grid points for global coverage. Parallel Message Passing Interface (MPI)
computing is used to increase computational efficiency.

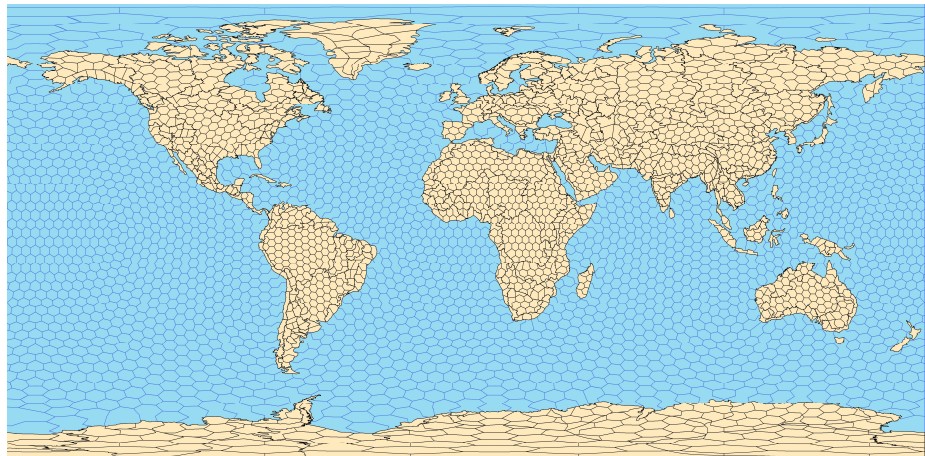

**Figure 1. Mascon partitioning of GCL-Mascon2024 solution**

### 2.3 Background Force Models and Input Data

Using the aforementioned methodology and mascon partitioning strategy, we have produced a time series of mascon solutions
(GCL-Mascon2024) from the GRACE Level-1B data covering the time period from January 2003 to December 2012. Here,
we concisely introduce the background force models and input tracking data.

Table 1 provides an overview of the background force models, which encompass various components, including Earth's static
gravity field, third-body attractions, solid Earth (pole) tides, ocean (pole) tides, atmospheric tides, atmospheric and oceanic
dealiasing effects, and general relativistic correction. In addition to the background force models mentioned above, mention
that these additional force models and corrections are discussed in detail below. (i) elastic response of the solid Earth to mass
transport at the Earth's surface, (ii) Glacial Isostatic Adjustments (GIA), (iii) Earth ellipsoidal corrections, and (iv) low-degree
term corrections.

Table 1 also lists the input used in mascon recovery, including nongravitational accelerations, satellite attitudes, reduced-
dynamic orbits, kinematic orbits, and k-band range-rate measurements. The Level-1B data used in the mascon recovery are
mainly from JPL, e.g., ACC1B, SCA1B, GNV1B, and KBR1B. Additionally, the kinematic orbit product released by the Graz
University of Technology (Strasser et al., 2019) was used in the GCL-Mascon2024 recovery framework.

**Table 1. Summary of Background Force Models and Data Used in GRACE Mascon Recovery**

|  | *GSFC RL06 mascon* | *CSR RL06 mascon* | *JPL RL06 mascon* | *GCL-Mascon2024* |
|---|---|---|---|---|
| **Background Force Model** | | | | |
| Static Earth Gravity | GGM05C | GGM05C (d/o 360) | GIF48 (d/o 180) | GOCO06s (d/o 300) (Kvas et al., 2021) |
| Solid Earth tides | IERS2010 conventions | IERS2010 conventions | IERS2010 conventions | IERS2010 conventions |
| Ocean tides | GOT4.7 (d/o 90) | GOT4.8 (d/o 180) | GOT4.7 and self-consistent equilibrium long-period tide (Convolution formalism to degree/order 90) | FES2014b (Lyard et al., 2021) |
| Solid pole tide | IERS2010 conventions (mean polar motion) | IERS2010 conventions (mean polar motion) | IERS2010 conventions (mean polar motion) | IERS 2010 conventions (mean polar motion) |
| Ocean pole tide | IERS2010 conventions | Desai models | IERS2010 conventions | Desai models (Desai, 2002) |
| Nontidal atmosphere and ocean dealiasing | ECMWF/MOG2D (Carrère and Lyard, 2003) | AOD1B RL06 | AOD1B RL06 | AOD1B RL06 (Dobslaw et al., 2017) |
| Atmospheric tides | - | - | - | AOD1B RL06 |
| Third-body attractions | * | DE-430 | DE-421 | DE-421 |
|---|---|---|---|---|
| General relativity | * | IERS2010 conventions | Point mass perturbation, geodesic and Lense-Thirring (Sun and Earth) | IERS2010 conventions |
| Nongravitational forces | 5s accelerometer data from GRACE Level-1B product | 5s accelerometer data from GRACE Level-1B product | 5s accelerometer data from GRACE Level-1B product | 5s accelerometer data from GRACE Level-1B product |
| **Local Parameters Estimated** | | | | |
| Satellite state | Position and velocity (Daily) | Position and velocity (Daily) | Position and velocity (Daily) | Position (2-hr) |
| GPS phase bias | * | - | Constant (Each GPS-GRACE pass) | - |
| KBR range-rate biases | Constant, drift, and once per revolution (3-hr) | - | Constant, drift, and once per revolution (One orbital revolution, 5400s) | - |
| Accelerometer | Bias | X, Y, and Z components (1.5-hr) | Along-track: 1/day linear Cross-track: 8/day linear Radial: 1/day linear | X, Y, and Z components (Daily) | X, Y, and Z components (2-hr) |
| | Drift | - | - | - | X, Y, and Z components (2-hr) |
| | Scale | - | Full matrix (Daily) | X and Y components (Monthly) | X, Y, and Z components (Daily) |
| | 1 cycle-per-revolution | 1.5-hourly 3-D one cycle-per-revolution empirical accelerations | - | - | - |
| **Satellite Observations** | | | | |
| Accelerometer observations | ACC1B RL02 with 1s sampling rate | ACC1B RL02 with 1s sampling rate | ACC1B RL02 with 1s sampling rate | ACC1B RL02 with 1s sampling rate |
| Attitude observations | SCA1B RL03 with 1s sampling rate | SCA1B RL03 with 1s sampling rate | SCA1B RL03 with 1s sampling rate | SCA1B RL03 with 1s sampling rate |
| GPS data | GPS1B RL03 with 30s sampling rate | GPS1B RL03 with 30s sampling rate | GPS1B RL03 with 30s sampling rate | - |
| Reduced-dynamic orbit | - | - | - | GNV1B RL02 with 5s sampling rate |
| Kinematic orbit | - | - | - | Kinematic orbits from Graz University of Technology with 10s sampling rate |
| K-/Ka Band Ranging satellite-to-satellite tracking measurement | KBR1B RL03 with 5s sampling rate | KBR1B RL03 with 5s sampling rate | KBR1B RL03 with 5s sampling rate | KBR1B RL03 with 5s sampling rate |
| **Details of Mascon Recovery** | | | | |
| Inversion approach | Dynamic approach | Dynamic approach | Dynamic approach | Short-arc approach |
| Inter-satellite observation | Range-rate | Range-rate | Range-rate | Range-rate |
| Satellite observations | Level-1B | Level-1B | Level-1B | Level-1B |
| Mascon count | 41168 | 40962 | 4551 | 4096 |
| Mascon shape (native resolution) | 1-arc-degree equal-area cells | 1-degree equal-area geodesic grid | 3-degree equal-area spherical cap | Land mascon ~ 300×300 km, ocean mascon ~ 400×400 km, and variable-shaped geometry constrained to coastlines and basin boundaries |
| Product resampled resolution | 0.5°×0.5° | 0.25°×0.25° | 0.5°×0.5° | 1.0°×1.0° |
| The relationship between inter-satellite | The mascons are related to the inter-satellite measurements via a | The mascons are related to the inter-satellite measurements via a | The mascons are related to the inter-satellite measurements via the | The mascons are related to the inter-satellite measurements via the |



| measurements and mascons | spherical harmonic expansion that is truncated at a finite degree and order. | spherical harmonic expansion that is truncated at a finite degree and order. | explicit partial derivatives with analytical expression. | explicit partial derivatives with analytical expression. |
|---|---|---|---|---|
| **Other Corrections** | | | | |
| Glacial Isostatic Adjustment Corrections | ICE6G-D (Peltier et al., 2015) | ICE-5G (Geruo et al., 2013) | Paulson model (Paulson et al., 2007) and ICE-5G (Peltier, 2004) | ICE6G-D (Stuhne and Peltier, 2015) |
| Low-degree Term corrections | Degree-1 terms replaced using Sun et al. (2016). $C_{20}$ replaced by TN-14 (Loomis et al., 2020). | Degree-1 terms replaced using Swenson et al. (2008). $C_{20}$ replaced with an SLR-derived value(Cheng and Tapley, 2004). | Degree-1 terms replaced using Swenson et al. (2008). $C_{20}$ replaced with an SLR-derived value (Cheng and Tapley, 2004). | Degree-1 terms replaced using Sun et al. (2016). $C_{20}$ replaced with an SLR-derived value (Cheng et al., 2011). |
| Earth Ellipsoidal correction | - | Ellipsoidal corrections from Ditmar (2018) | Ellipsoidal corrections from Li et al. (2017) | Ellipsoidal corrections from Ditmar (2018) |
| Mean removed | 2004.0-2010.0 | 2004.0-2010.0 | 2004.0-2010.0 | 2004.0-2010.0 |

\* Data missing; - Data or strategy unavailable

### 2.3.1 Earth's Elastic Response

The solid Earth is not perfectly rigid but exhibits some elastic response to surface loads (Boy and Chao, 2005). Here, we estimate the effect of surface load or surface mass changes based on the elastic loading theory of a spherical Maxwell Earth, according to Wahr et al. (1998), who used load Love numbers (represented as $k_l$) to quantify Earth's elastic deformation.

In this study, the temporal gravity field model released by the Institute of Geodesy of the Graz University of Technology (ITSG-Grace2018 (Kvas et al., 2019)) is used as the signal source to compute the Earth's elastic deformation. Because this

model is represented in terms of unfiltered spherical harmonic coefficients, there exists north-south stripes and high-frequency noise in the spatial domain. Thus, postprocessing in the form of the DDK4 filter (Kusche et al., 2009) is used to mitigate these issues. The elastic deformations induced by the filtered ITSG-Grace2018 solutions are incorporated into the GCL-Mascon2024 recovery framework as an additional background force model.

### 2.3.2 Glacial Isostatic Adjustments

We apply GIA corrections in the GCL-Mascon2024 recovery process as another background force model. The official mascon products (i.e., CSR RL06 mascon, JPL RL06 mascon, and GSFC RL06 mascon) represent the surface mass deviation relative to the 2004.0-2009.999 time-mean baseline. Subsequently, we model the GIA signals relative to the middle epoch of 2007.000, utilizing the GIA model ICE-6G, which was developed by Stuhne and Peltier (2015).

### 2.3.3 Earth Ellipsoidal Corrections

Temporal Stokes coefficients derived from GRACE satellite data are typically converted into mass anomalies at the Earth's surface using spherical harmonic synthesis, as formulated by Wahr et al. (1998). However, the results obtained using this approach reflect mass transport at a spherical surface with a fixed radius of 6378 km, which can introduce inaccuracies. Ditmar (2018) demonstrated that such a conversion may lack sufficient precision and proposed a revised formulation for converting Stokes coefficients into mass anomalies. This updated approach assumes that: (i) mass transport occurs at the reference

ellipsoid, and (ii) at each point of interest, the ellipsoidal surface is approximated by a sphere with a radius equal to the local radial distance from the Earth's center (the "locally spherical approximation"). In this study, we adopt the spherical harmonic synthesis method proposed by Ditmar (2018) to account for the effects of the Earth's oblateness and improve the accuracy of mass anomaly estimation.

### 2.3.4 Low-degree Term Corrections

Given the inherent limitations of the GRACE twin-satellite tracking, it is not feasible to determine the effects of geocenter motion, which can be represented in terms of time-varying degree-1 coefficients. Consequently, we utilize the coefficients derived by combining GRACE data with geophysical models (Sun et al., 2016). Furthermore, we incorporate the $C_{20}$ (degree 2 order 0) coefficients derived from Satellite Laser Ranging (SLR) measurements (Chen et al., 2005; Cheng et al., 2013) to enhance accuracy. To this end, and in line with previous studies (Watkins et al., 2015), the mascon grid solutions are first

converted to the spherical harmonic coefficients by using spherical harmonic analysis. Then, we replace the low-degree terms (i.e., degree-1 and $C_{20}$) and utilize spherical harmonic synthesis proposed by Ditmar (2018); the coefficients are converted back to mascon grid solutions to correct the implied low-degree term component of GCL-Mascon2024, considering the influence of the Earth's oblateness as detailed in section 2.3.3.

### 2.4 Frequency-Dependent Data Weighting

The concept of FDDW originates from the fast collocation technique (Bottoni and Barzaghi, 1993), which assumes stationary measurement noise, thereby imparting a Toeplitz structure to the noise covariance matrix. Subsequently, Ditmar et al. (2007) provided a detailed discussion of the FDDW concept and employed the technique to estimate the static Earth gravity field from the kinematic orbital acceleration of the CHAllenging Minisatellite Payload (CHAMP) satellite (Ditmar et al., 2006). The FDDW technique was later adapted for solving the temporal gravity field model using the GRACE inter-satellite acceleration

(Liu et al., 2010). Afterward, Guo et al. (2018) utilized the FDDW technique to account for KBR frequency-dependent noise in the classical dynamic approach, leading to the development of the WHU RL01 model. Chen et al. (2019) further extended the application of the FDDW technique by incorporating both orbit and KBR frequency-dependent noise into the optimized short-arc approach and released the temporal gravity model named the Tongji-Grace2018 solution.

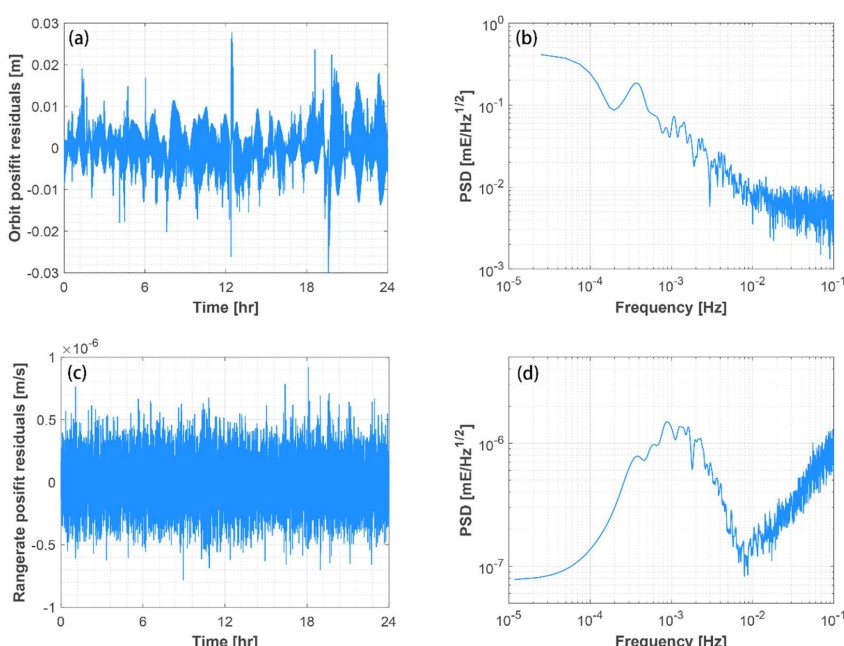

**Figure 2. Time series and power spectrum densities (PSD) of postfit residuals from orbit and KBR range rate**

As indicated in numerous previous studies (e.g., Guo et al., 2018; Chen et al., 2019), the inter-satellite range-rate measurements are affected by frequency-dependent noise. Before applying the FDDW technique, it is essential to build a stochastic noise

model using, e.g., postfit residuals from the GRACE measurements. As an example, we select the postfit residuals from June

2009, calculated using the preliminary mascon solution for that month. As shown in Figure 2 (a) and (c), the time series of

postfit residuals from orbit and range-rate measurements on 5 June 2009, exhibit a clear dependence on frequency. This is

further illustrated by the power spectral densities (PSDs) displayed in Figure 2 (b) and (d), which indicate that both orbit and

range-rate measurements, particularly the former ones, are contaminated by low-frequency noise. The frequency-dependent

noise in GRACE observations is largely attributed to errors in the GRACE orbits (Ditmar et al., 2012). This type of noise, in

the essence of perfect orbital/instrumental/other models, is typically addressed by either estimating (once or twice per orbital

revolution) periodic parameters to account for unmodeled accelerations or incorporating variance-covariance matrices to

mitigate these errors (Zhou et al., 2024). In this study, noise whitening filters $W$, constructed based on postfit residuals derived

from orbit and range-rate measurements using the autoregressive (AR) noise model implemented in the ARMASA toolbox

(Broersen and Wensink, 1998; Broersen, 2000), are applied to transform frequency noise $\varepsilon$ into Gaussian white noise $\hat{\varepsilon}$.

Following the methodology of Chen et al. (2019), the variance-covariance matrix $\Sigma$ can be constructed using the law of

variance-covariance propagation:

$$\Sigma = W^{-1} \cdot diag\left[\left(W \cdot \varepsilon\right)^2 \Big/ \left(\varepsilon_0\right)^2\right] \cdot \left(W^T\right)^{-1} . \tag{4}$$

## 2.5 Advanced Spatial Constraints

The linear system that connects satellite range-rate observations to the mass anomalies within each mascon for estimation is

rank-deficient. To stabilize the rank-deficient system of equations in mascon recovery, we employ Tikhonov regularization

techniques (Tikhonov, 1963). Herein, we estimate the mascon elements using the following equation:

$$\hat{x} = \left(A^T P A + C_M\right)^{-1} \cdot A^T P L , \tag{5}$$

where $\hat{x}$ represents the mascons to be estimated; $A$ is the design matrix of partial derivatives; $L$ is the residual vector which is

obtained by subtracting the kinematic orbit or KBR measurements from the reference orbit positions or KBR data; $P$ is the

weight matrix derived from the inverse of the variance-covariance matrix $\Sigma$ (refer to section 2.4); $C_M$ is a diagonal constraint

(or regularization) matrix of size $n \times n$, named the Mass Variation Regularization Constraint Normalized (MVRCN) Matrix; $n$

is the number of the mascons to be estimated.

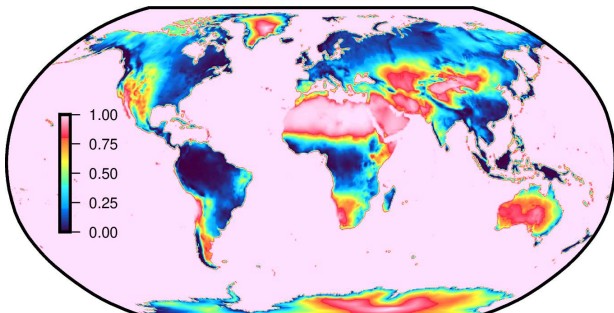

**Figure 3. The Mass Variation Regularization Constraint Normalized (MVRCN) Matrix**
**used in the GCL-Mascon2024 recovery framework**

For the advanced spatial constraints, we construct the MVRCN matrix, which primarily comprises two components: one

derived from the continental region aridity-wetness index, which is defined as the ratio of mean annual precipitation to mean

annual reference evapotranspiration (Trabucco and Zomer, 2018), and the other from the ETOPO Global Relief Model of ice

sheet regions (i.e., Greenland and Antarctica) in ice surface version that portrays the topography of the top layer of the polar

ice sheets (Macferrin et al., 2024). The fundamental premise is that humid basins on the continent require looser constraints

for recovering higher temporal gravity signals, while arid basins require tighter constraints. Similarly, in polar ice sheets, areas at lower elevations necessitate looser constraints to recover mass variations, whereas regions at higher elevations require tighter constraints. Figure 3 shows the spatial distribution of the MVRCN Matrix. The previously described MVRCN Matrix is then tuned to an appropriate value using the L-curve method, ensuring that the resulting regularization matrix is sufficiently tight to suppress noise yet loose enough to allow the mascons to adjust to their optimal values.

## 3 Short-Arc Approach for Gravity Field Inversion


The short-arc approach, initially introduced by Schneider (1968), is a commonly utilized method for satellite gravity data inversion. Mayer-Gürr (2008) further proposed a gradient correction algorithm to enhance the accuracy of the short-arc approach and applied it to real GRACE data inversion. Since then, the short-arc approach has been employed in processing GRACE data (Ran et al., 2014; Chen et al., 2019), demonstrating its effectiveness and efficiency in recovering temporal gravity

field models. Section 3.1 is devoted to the optimal choice of the arc length. Next, section 3.2 discusses the design of calibration parameters estimation during the gravity inversion process.

### 3.1 Arc Length Determination

Longer arcs (e.g., 24-hr ones) are usually utilized in the dynamic approach to the temporal gravity solution recovery, whether it be in the form of the mascon solution (e.g., Watkins et al., 2015) or spherical harmonic solutions (e.g., Mayer-Gürr et al.,

2018). Regarding the short-arc approach, the tendency is to select shorter arc lengths, such as 1-hr arcs for Bonn University's ITG-GRACE2010 (Mayer-Guerr et al., 2010) and 6-hr arcs for Tongji University's Tongji-Grace2018 (Chen et al., 2019). However, as the arc length decreases, the number of parameters per day increases. Given that the total number of observations remains constant, this increases the condition number to the estimation process in the temporal gravity field recovery. In the mathematical sense, the smaller the condition number of the normal matrix, the more stable the resulting estimate of the gravity

field (Chen et al., 2019).

To determine the appropriate arc length for GCL-Mascon2024, we conducted computations of a monthly mascon model using different arc lengths to compare the stability of the resulting estimates. Table 2 presents the condition numbers of the unconstrained normal matrices and the corresponding computational time needed for different arc lengths. From this standpoint, the 2-hr arc length corresponds to the most stable arc length in the GCL-Mascon2024 recovery. Figure 4 illustrates that

increasing the arc length beyond 2 hr in the short-arc approach leads to a significant increase in noise in gravity field estimates as the normal equations become more ill-conditioned. This observation aligns closely with what we conclude from Table 2. Therefore, an arc length of 2-hr is determined to be the most suitable for the short-arc approach employed in this work.

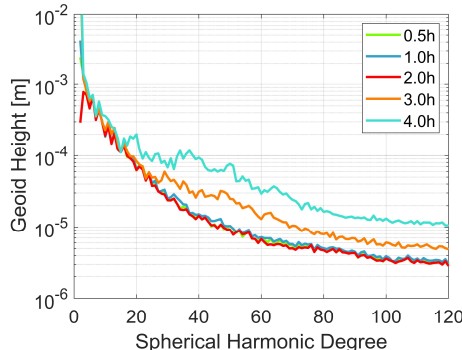

**Figure 4. Geoid height differences per degree w.r.t GOCO06s from mascon solutions of different arc lengths**






**Table 2. Condition numbers (log10) of normal matrixes and inversion time cost in GCL-Mascon2024 recovery framework with different arc lengths**

| Arc Length/hr | Condition Numbers/log10 | Time Cost/hr |
|---|---|---|
| 0.5 | 8.41 | 6.69 |
| 1.0 | 7.95 | 9.04 |
| 2.0 | 7.93 | 16.82 |
| 3.0 | 7.95 | 29.08 |
| 4.0 | 7.99 | 47.16 |
| 6.0 | 8.28 | 70.78 |

### 3.2 Calibration Parameters Estimation

The accelerometer represents a significant source of errors in the GRACE mission (Kim, 2000), necessitating the
implementation of robust strategies to manage and mitigate accelerometer errors effectively. Simultaneously, in the analysis
of GRACE observations, it is necessary to estimate not only the gravity field parameters but also arc-related parameters, such
as the two boundary position vectors of each arc (Mayer-Gürr, 2008). That is, the error occurring at the boundaries of each arc
is also of non-negligible magnitude. A commonly used strategy in temporal gravity field recovery is the incorporation of
calibration parameters to mitigate the impact of the aforementioned errors.

The GRACE raw accelerometer measurements exhibit systematic errors, including bias, scale error, and drifts (Han et al.,
2006b) in three axes (i.e., along, cross, and radial) for both satellites. The findings of Meyer et al. (2016) demonstrate that the
scale calibration of accelerometer data at daily intervals significantly reduces the impact of solar activity on the derived gravity
field models. To this end, we conduct the daily estimation of accelerometer scales in three axes for both satellites in this study.
In addition, bias is a frequently employed parameter for estimating the local parameters of accelerometers (Kim, 2000). Based
on prior studies (e.g., Han et al., 2006b; Bettadpur, 2007), we also incorporate the estimation of drift parameters into the
recovery of the mascon solution. Combining the biases, drifts, and scales, the calibration formula for the accelerometer data
can be constructed as

$$f_{new} = bias + scale \times f_{ori} + drift \times t \, , \tag{6}$$

where $f_{ori}$ and $f_{new}$ denote the nongravitational accelerations prior to and after calibration, respectively; **bias**, **scale**, and **drift**
are the estimated local parameters of the accelerometers; $t$ represents the period about which the drift of nongravitational
accelerations is calibrated. Figure 5 illustrates the geoid height differences per degree with respect to GOCO06s of the mascon
solutions with accelerometer calibration parameters (i.e., bias, drift, and scale) co-estimated over different periods.

**Table 3. Estimation periods of accelerometer calibration parameters (Unit: minutes)**

| Case | Bias | Drift | Scale |
|---|---|---|---|
| A | 120 | 120 | 1440 |
| B | 120 | 360 | 1440 |
| C | 120 | 720 | 1440 |
| D | 120 | 1440 | 1440 |
| E | 360 | 360 | 1440 |
| F | 360 | 720 | 1440 |
| G | 360 | 1440 | 1440 |
| H | 720 | 720 | 1440 |
| I | 720 | 1440 | 1440 |
| J | 1440 | 1440 | 1440 |

Table 3 provides a detailed definition of each considered case, characterized by three pre-defined periods for accelerometer calibration parameters: bias, drift, and scale. One can see from Figure 5 that the inversion performs optimally when bias and drift are co-eliminated per arc as well as scale elimination on a per-day basis with the premise of estimating the boundary position parameters per arc. After generating the normal equation for each arc, the calibration parameters of the boundary position can be eliminated immediately. Then, once the normal equations for a specific period are generated, the corresponding

accelerometer calibration parameters are eliminated as well. Last, by combining all the reduced daily normal equations, we obtain the final monthly normal equation, which is solved for the mascon coefficients.

As mentioned above, a 2-hr arc is selected for the GCL-Mascon2024 computation. The calibration parameters for accelerometer observations include biases and drifts estimated per arc, as well as scales estimated per day for twin-satellite in three axes.

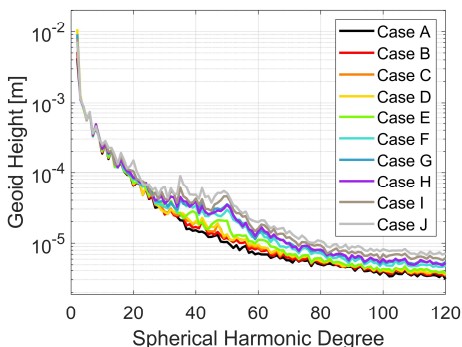


**Figure 5. Geoid height differences per degree w.r.t GOCO06s from mascon solutions under different scenarios. Each scenario corresponds to a distinct set of parameters, reflecting variations in the estimation periods for accelerometer calibration parameters (i.e., bias, drift, and scale). Refer to Table 3 for detailed information on parameter settings.**

**4 Analysis of Scientific Results**

To evaluate and validate the GCL-Mascon2024 solution, we compare the estimates of mass variation globally and over specific regions with the RL06 mascon solutions released by GSFC, CSR, and JPL. Here, annual amplitudes, monthly mass variations, basin hydrological signals, and polar region mass balance are utilized to assess the performance of temporal signal retrieval. At the same time, continental random noise and desert residuals are used to evaluate temporal noise.

**4.1 Global Comparisons**

We first analyze the global mass change signals in GCL-Mascon2024 and in the RL06 mascon solutions provided by GSFC, CSR, and JPL. To emphasize the differences in the four mascon solutions, the results are presented as anomalies in relation to the baseline defined as the time-mean in the period from January 2004 to December 2009. In Figure 6, we specifically present the temporal gravity signals in April 2008. Upon observing Figure 6, we can discern a high level of consistency in the global mass change signals across all four models.

The annual amplitude in total water storage is depicted in Figure 7 for the time span ranging from January 2003 to December 2012. It is evident that the spatial distribution of the four monthly mascon solutions exhibits a substantial level of concurrence. Regions characterized by a more pronounced annual fluctuation in total water storage predominantly concentrate in specific areas, namely the Amazon basin in South America, the Niger basin in West Africa, the Zambezi basin in South Africa, as well as the Ganges and Mekong basins in Southeast Asia.




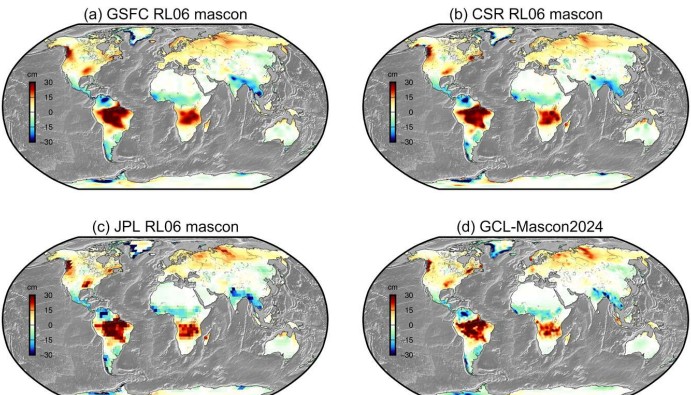

**Figure 6. Global mass change signals in April 2008 derived from the GCL-Mascon2024 and the RL06 mascon solutions offered by the GSFC, CSR, and JPL. The reference is the average model over the period from January 2004 to December 2009 (in equivalent water height, or EWH).**

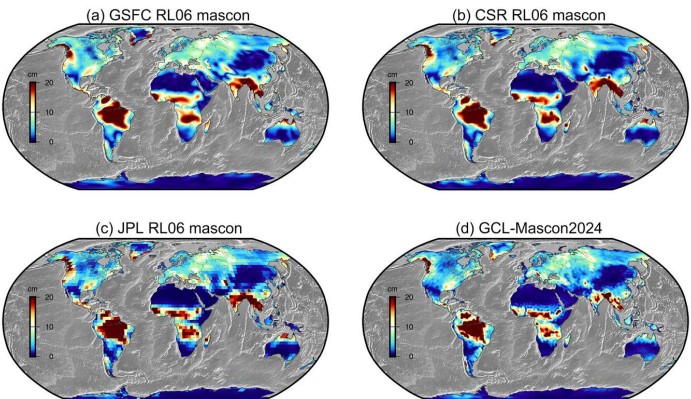


**Figure 7. Annual amplitudes of total water storage mass change from the GCL-Mascon2024 and the RL06 mascon solutions offered by the GSFC, CSR, and JPL (in equivalent water height, or EWH).**

Ditmar (2022) proposed a technique to combine and regularize GRACE-based mass-anomaly time series and, at the same time, to quantify the Standard Deviation (SD) of random noise in each time series. The latter is estimated using Variance Component

Estimation (VCE) as adapted from Koch and Kusche (2002). Figure 8 illustrates the spatial distribution of the random noise SD estimated for various mascon solutions. The noise SD of the mass-anomaly time series over the globe obtained for the mascon solutions from GSFC, CSR, and JPL, along with the GCL-Mascon2024, are 4.6 cm, 5.4 cm, 5.4 cm, and 4.3 cm, respectively. In northern Africa, the Arabian Peninsula, and eastern Asia (the border region between China and Mongolia), GCL-Mascon2024 and JPL mascon solutions exhibit similar spatial distributions with smaller SD of random noise compared

to GSFC and CSR solutions. Given the predominant desert coverage in these regions, it is reasonable to expect lower standard deviations of random noise. Further quantitative analyses of random noise over specific local regions, including river basins, Greenland, and desert areas, are provided in the following section. Figure 8 shows that the GCL-Mascon2024 can reduce the error by 6.5%−20.4% compared to the RL06 mascon solutions produced by GSFC, CSR, and JPL.

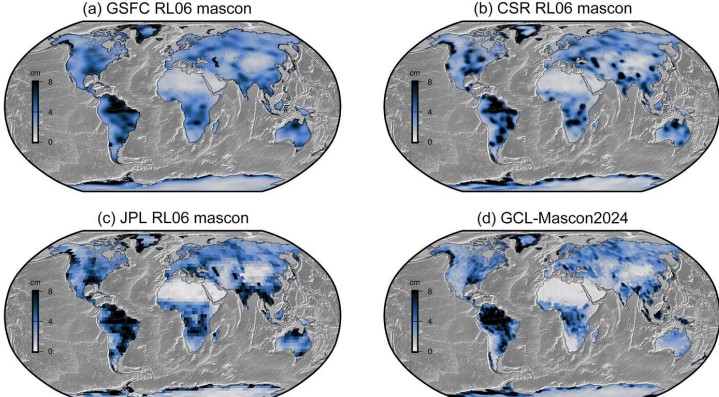

**Figure 8. Spatial distribution characteristics of random noise of GSFC RL06 mascon, CSR RL06 mascon, JPL RL06 mascon, and GCL-Mascon2024 (in equivalent water height, or EWH), with the standard deviation computed according to Ditmar (2022).**

### 4.2 Regional Comparisons

For a more comprehensive comparative analysis of signal magnitudes across various mascon solutions, this study selects distinct river basins, Greenland drainage systems, and typical deserts. These specific selections allow us to discern temporal
signals associated with hydrological processes, ice melting dynamics, and temporal noise, respectively.

### 4.2.1 Total Water Storage in Hydrology

Continental water storage is a pivotal constituent within both terrestrial and global hydrological cycles, exerting a significant degree of control over intricate processes involving water, energy, and biogeochemical exchanges (Famiglietti, 2003). As such, it plays a paramount role in shaping and influencing the Earth's climate system (Chen et al., 2010). Of significant importance
in terrestrial basins, the comprehensive analysis of Total Water Storage (TWS) aids in understanding the intricate dynamics of water distribution and availability (Long et al., 2013). TWS refers to the summation of all water present within a given region, accounting for its various forms, such as surface water, groundwater, soil moisture, and snowpack. The GRACE mission can accurately capture the total mass variation caused by terrestrial water storage change (e.g., Ramillien et al., 2008; Rodell et al., 2018).

Given the potential divergence in temporal signals of mass variations across river basins characterized by distinct sizes and climate classifications, we have statistically analyzed the temporal signals within the 42 largest basins (area > $5{\times}10^5$ km$^2$) in the world, which encompasses different climate types. This selection intends to showcase the performance of the temporal signals recovery by the different mascon solutions. The basic definitions of the aforementioned river basins are all taken from and credited to Scanlon et al. (2018).

**Table 4. Correlation coefficients between mass anomaly time series over the representative river basins from the GCL-Mascon2024 and from official RL06 Mascon solutions**

| Basin name | GSFC | CSR | JPL |
|---|---|---|---|
| Amazon | 0.9985 | 0.9986 | 0.9986 |
| Mississippi | 0.9848 | 0.9874 | 0.9844 |
| Ob | 0.9887 | 0.9923 | 0.9920 |
| Lena | 0.9813 | 0.9818 | 0.9799 |
| Murray | 0.9649 | 0.9499 | 0.9427 |
| Orinoco | 0.9926 | 0.9944 | 0.9940 |
| Tocantins | 0.9882 | 0.9872 | 0.9844 |
| Mekong | 0.9922 | 0.9933 | 0.9899 |

Figure 9 illustrates the time series of basin mass variations derived from the WaterGAP Global Hydrology Model (WGHM), GCL-Mascon2024, and the mascon solutions from GSFC, CSR, and JPL, respectively. The WaterGAP model (Schmied et al., 2021; Müller Schmied et al., 2023), primarily developed at the Universities of Kassel and Frankfurt, simulates water flows,

storage, withdrawals, and consumptive use globally, serving as a tool to evaluate the human–freshwater system under the influence of global change. As shown in Figure 9, GCL-Mascon2024 exhibits a high level of agreement with the other models in terms of mass anomalies across all analyzed river basins. Using WGHM-based mass variations as control data, the time series derived from GCL-Mascon2024 for the 42 largest basins demonstrates an approximately 8.7% reduction in error compared to the other three mascon solutions released by GSFC, CSR, and JPL, respectively. Notably, in the Murray Basin,

which exhibits the sub-arid climate type, the GCL-Mascon2024 time series shows a 48.2%−64.4% reduction in error compared to the other mascon solutions. As shown in Table 4, the correlation coefficients for mass variations within the selected regions between GCL-Mascon2024 and the other mascon solutions exceed 95.0%.

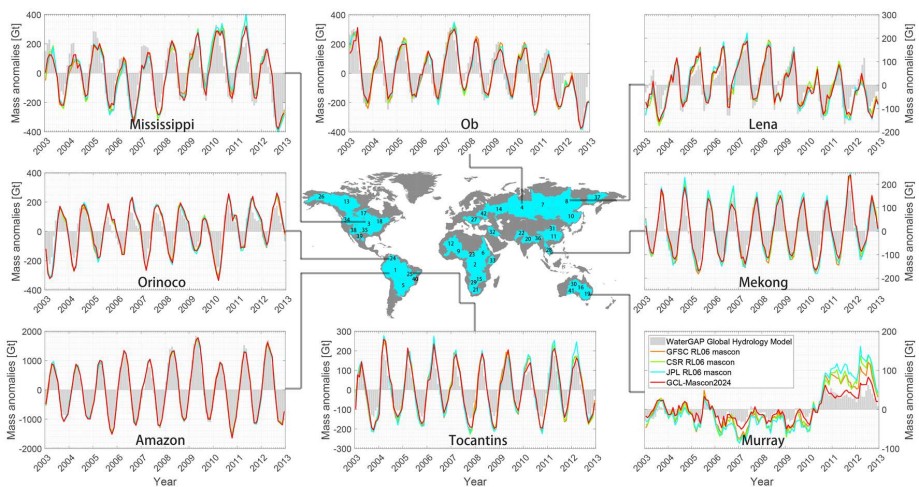

**Figure 9. Time series of mass anomalies over typical river basins from the hydrology model WaterGAP (outlined by the grey zone)**
**and mascon solutions recovered by GSFC, CSR, JPL, and GCL (yellow, green, blue, and red lines, respectively). The base map illustrates the 42 largest basins (area > 5 × 10⁵ km²) extracted from the Total Runoff Integrating Pathway database as from Scanlon et al. (2018).**

According to Table 5, the noise SD of the mass-anomaly time series over the aforementioned river basins for the mascon solutions from GSFC, CSR, and JPL, along with the GCL-Mascon2024, are 3.9 cm, 4.3 cm, 4.6 cm, and 3.6 cm, respectively.

It is important to highlight that the ability of the GCL-Mascon2024 solution to suppress random noise is optimal in non-humid (i.e., subhumid, semiarid, and arid) basins. This indicates that the noise reduction of the GCL-Mascon2024 solution is 32.5%, 37.7%, and 40.0%, respectively, compared to the GSFC, CSR, and JPL RL06 mascon solutions. Those improvements provided by the GCL-Mascon2024 solution may benefit from incorporating advanced spatial constraints derived from the aridity-wetness index of continental regions.

The results presented in Figure 9, Table 4, and Table 5 demonstrate strong evidence that GCL-Mascon2024 is equally sensitive to hydrological signals as the official mascon solutions despite employing a shorter arc length (i.e., 2 hr) and exhibits a superior capacity for random noise suppression.




**Table 5. The root mean square of random noise over the 42 largest basins (area > 5 × 10⁵ km²) from the mascon solutions recovered by GSFC, CSR, JPL, and GCL. The definitions of these basin boundaries are derived from Scanlon et al. (2018). The bolded value indicates the lowest RMS of random noise. (Unit: centimeters)**

| ID | Basin name | Climate type | GSFC | CSR | JPL | GCL |
|----|------------|--------------|------|-----|-----|-----|
| 1 | Amazon | Humid | **7.3262** | 8.1324 | 8.7691 | 9.9309 |
| 2 | Congo | Humid | 4.5035 | **4.3349** | 5.2680 | 5.0626 |
| 3 | Mississippi | Humid | 4.7503 | 4.9555 | 5.5488 | **4.2651** |
| 4 | Ob | Humid | 3.7164 | 3.7100 | 3.8463 | **3.3907** |
| 5 | Parana | Humid | **5.3869** | 6.4401 | 6.6431 | 5.5095 |
| 6 | Nile | Semiarid | 3.0908 | 3.9795 | 4.3371 | **2.6374** |
| 7 | Yenisei | Humid | 3.5998 | **3.5924** | 3.8672 | 3.7469 |
| 8 | Lena | Humid | 3.0510 | **2.9814** | 3.3214 | 3.0428 |
| 9 | Niger | Semiarid | 2.2399 | 2.3946 | 2.7950 | **2.1589** |
| 10 | Amur | Humid | 3.5337 | 3.4579 | 3.5756 | **3.3445** |
| 11 | Yangtze | Humid | 3.5906 | **3.5156** | 4.2037 | 4.1323 |
| 12 | Tamanrasset | Arid | 1.3854 | 1.0377 | 0.7405 | **0.7118** |
| 13 | Mackenzie | Humid | 2.7168 | 2.3801 | 2.7205 | **2.3514** |
| 14 | Volga | Humid | 4.3060 | **4.0066** | 4.6988 | 4.2466 |
| 15 | Zambezi | Subhumid | 5.6295 | 6.7237 | 6.7917 | **4.2623** |
| 16 | Lake Eyre | Arid | 4.0057 | 3.4419 | 3.5115 | **2.3293** |
| 17 | Nelson | Humid | 4.6281 | 4.6112 | 5.2376 | **4.1103** |
| 18 | St. Lawrence | Humid | 4.2487 | 5.3319 | **4.2335** | 4.3893 |
| 19 | Murray | Semiarid | 4.1303 | 4.4812 | 5.0861 | **2.6322** |
| 20 | Ganges | Humid | **5.0676** | 8.1039 | 7.2558 | 5.1471 |
| 21 | Orange | Semiarid | 2.2863 | 2.1591 | 2.2189 | **0.9951** |
| 22 | Indus | Semiarid | 3.6739 | 3.8905 | 4.7001 | **2.6028** |
| 23 | Chari | Semiarid | 2.6988 | 2.4376 | 3.2448 | **1.9925** |
| 24 | Orinoco | Humid | **6.5169** | 7.4842 | 8.3259 | 8.9325 |
| 25 | Tocantins | Humid | **6.3024** | 7.2767 | 8.1663 | 7.6079 |
| 26 | Yukon | Humid | 3.4357 | **3.2947** | 4.4142 | 3.5464 |
| 27 | Danube | Humid | **4.5834** | 5.0608 | 5.6181 | 5.2399 |
| 28 | Mekong | Humid | **4.9504** | 5.4990 | 7.3538 | 5.4697 |
| 29 | Okavango | Semiarid | 4.0300 | 4.7889 | 4.7980 | **1.9694** |
| 30 | Victoria | Arid | 5.7220 | 5.9436 | 5.7244 | **3.4492** |
| 31 | Huang He | Subhumid | 2.9258 | 3.5305 | 2.7982 | **1.9563** |
| 32 | Euphrates | Semiarid | 3.2190 | 3.9260 | 4.1723 | **1.8597** |
| 33 | Jubba | Semiarid | 2.4890 | 1.9113 | 2.1410 | **1.3877** |
| 34 | Columbia | Humid | 3.0176 | 2.8252 | 4.4262 | **2.7851** |
| 35 | Arkansas | Subhumid | 5.2859 | 5.8052 | 6.6062 | **4.9866** |
| 36 | Brahmaputra | Humid | **3.7799** | 5.5452 | 5.0203 | 4.7857 |
| 37 | Kolyma | Humid | 2.6953 | **2.3267** | 2.9074 | 2.5333 |
| 38 | Colorado | Semiarid | 2.7855 | 2.0176 | 2.6225 | **1.7487** |
| 39 | Rio Grande | Semiarid | 3.4130 | 2.8457 | 3.3305 | **2.0252** |
| 40 | Sao Francisco | Subhumid | 5.5532 | 9.1857 | 7.6804 | **4.0576** |
| 41 | Nullarbor | Arid | 2.8451 | 2.5436 | 2.4602 | **1.7504** |
| 42 | Dniepr | Humid | 3.8802 | 3.6757 | 3.9689 | **3.5107** |

**400  4.2.2 Mass Balance of Greenland Ice Sheet**

The Greenland Ice Sheet (GrIS) is home to one of the largest freshwater reserves on our planet. Due to its substantial accumulation rate and considerable meltwater runoff, the GrIS is a highly dynamic system (Chen et al., 2006). Rapid transformations within the GrIS have the potential to raise the mean sea level substantially (Ran et al., 2024) and could significantly impact the North Atlantic thermocline circulation, thereby affecting the global climate(Velicogna and Wahr,


2005). One of the primary means for monitoring mass variation in the GrIS is the GRACE satellite mission (e.g., Schlegel et al., 2016; Velicogna et al., 2020).

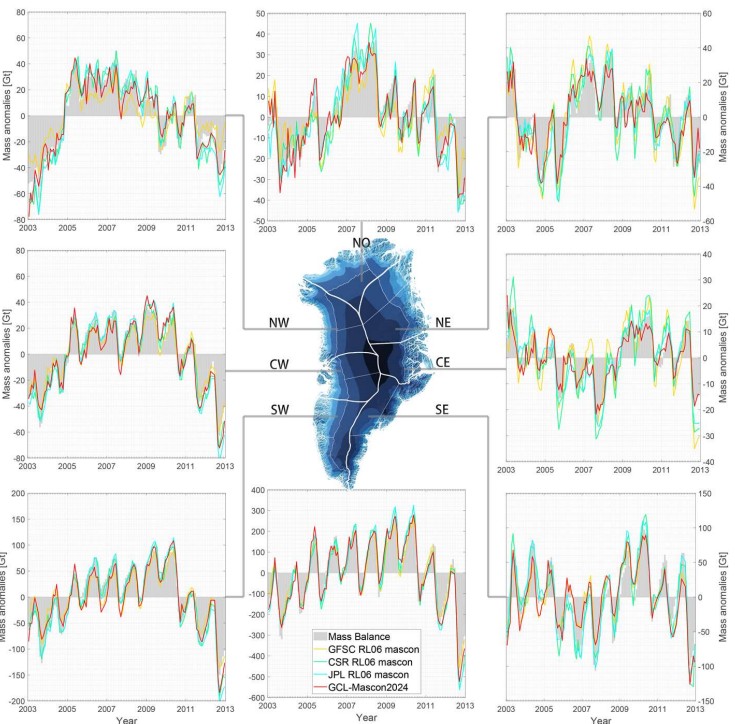

**Figure 10.** Time series of de-trended mass anomalies for individual drainage systems and the entire Greenland, based on the mass balance from the Input-Output Method, i.e., Surface Mass Balance – Ice Discharge (outlined by the grey zone) and mascon

solutions recovered by GSFC, CSR, JPL, and GCL (yellow, green, blue, and red lines, respectively). The middle panel presents the schematic illustration of the mascon division and its base map portrays the topography of the Greenland Ice Sheet. In this study, Greenland is partitioned into 21 mascons and 7 individual drainage systems: North (NO), Northeast (NE), Northwest (NW), Central East (CE), Central West (CW), Southeast (SE), and Southwest (SW).

In Greenland, it is critical to emphasize that the mascon geometry of GCL-Mascon2024 is delineated based on the boundaries

of the Greenland drainage system and the coastline. The Greenland is partitioned into 21 mascons and 7 individual drainage systems: North (NO), Northeast (NE), Northwest (NW), Central East (CE), Central West (CW), Southeast (SE), and Southwest (SW). The various mascon solutions over different drainage systems of Greenland are validated using the Input-Output Method (IOM) as control data, i.e., mass balance = Surface Mass Balance – Ice Discharge. Mass variations caused by surface mass balance (SMB) processes are derived from the MARv3.14.0 polar regional climate model run at a resolution of 10 km over

the whole GrIS and 6 hourly forced by the ERA5 reanalysis at its lateral boundaries and over the ocean (Fettweis et al., 2017). The middle panel of Figure 10 presents the schematic illustration of the mascon division and the topography of the ice surface on Greenland. The other subfigures of Figure 10 illustrate the time series of de-trended mass anomaly based on the mass balance from IOM outlined by the grey zone and the different mascon solutions integrated over 7 drainage systems, as well as over the entire Greenland. As indicated in Figure 10, the time series of mass changes over Greenland is generally consistent

across the four different mascon solutions, with all models effectively capturing the overall mass change in Greenland. The correlation coefficients of mass changes across the seven different drainage systems between GCL-Mascon2024 and the other three RL06 mascon solutions exceed 97.7%. Furthermore, the correlation coefficient for capturing the total mass change of Greenland across all four models is as high as 99.8%. Particularly in the North drainage system of Greenland, where the mass variation is minimal, the time series for this region, extracted from GCL-Mascon2024, demonstrates a 10.1%−40.5% reduction

in error compared to the other three mascon solutions from GSFC, CSR, and JPL, respectively. By extracting the noise SD of



the mass anomaly time series within the Greenland drainage system from various mascon solutions (Table 6), we find that the noise SD for the GCL-Mascon2024 and GSFC RL06 mascon solutions is 9.3 cm and 9.0 cm, respectively, whereas it is 13.8 cm and 13.2 cm for the CSR and JPL RL06 mascon solutions. This indicates that the GCL-Mascon2024 solution achieves a random noise reduction of 32.3% and 30.0% compared to the CSR and JPL RL06 mascon solutions. The observed

discrepancies and the improvement offered by our mascon solution could be attributed to differences in the definition of mascon geometry and the processing methodology.

### 4.2.3 Mascon Solution Validation over Deserts

The preceding two sections have delved into the signal characteristics exhibited by the GCL-Mascon2024 solution over river basins and Greenland. In this section, we aim to evaluate the uncertainties of our mascon solutions over deserts and compare

them with those of the other mascon solutions. Our impetus stems from an understanding that mass variations within desert regions are minor. The residuals, calculated after removing the climatological components (i.e., bias, trend, and amplitude) from the mass variations, can be regarded as mis-modeling signals or temporal noise that persist in the temporal gravity fields. Consequently, we analyze the error characteristics inherent to the mascon models over typical deserts, such as the Sahara Desert in Africa, the Taklamakan Desert in Asia, and the Atacama Desert in South America.

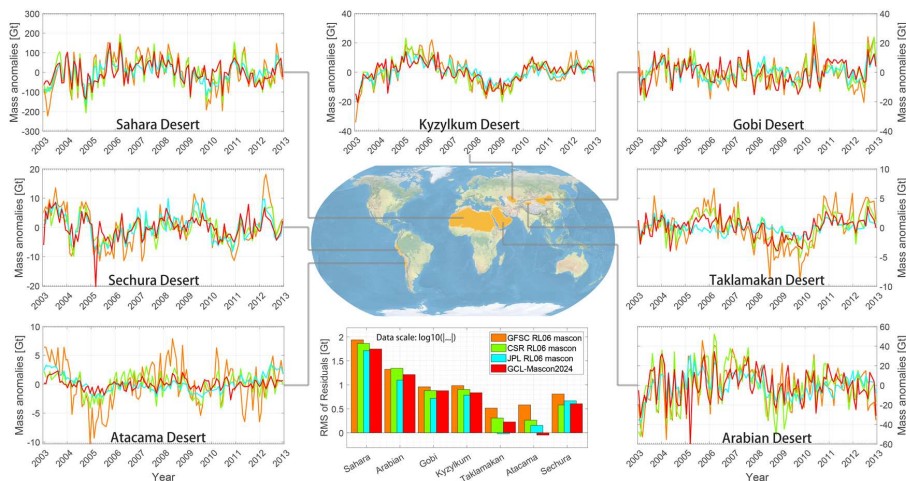


**Figure 11. Time series of mass change residuals over deserts derived from the RL06 mascon solutions from GSFC, CSR, JPL, and GCL-Mascon2024. The residuals indicate that the climatological components (i.e., bias, trend, and amplitude) have been removed from the mass variation. The deserts chosen are the Sahara Desert, Sechura Desert, Atacama Desert, Kyzylkum Desert, Gobi Desert, Taklamakan Desert, and Arabian Desert.**

Deserts are territories characterized by low precipitation. They can be classified into several categories based on their respective geographical locations and prevailing weather patterns, which include trade wind deserts, rain shadow deserts, and coastal deserts (Whitford and Duval, 2019). Trade wind deserts are typically found on both sides of the horse latitudes, between ±30° and ±35°. These regions are characterized by subtropical anticyclones and the large-scale descent of dry air masses (Glennie, 1987). The Sahara Desert, the largest hot desert in the world, is an example of this type. By extracting the mass

variation residuals of the Sahara Desert from varying mascon solutions, the residual of GCL-Mascon2024 solution and JPL RL06 mascon solution is 55.8 Gt and 51.7 Gt, but it is 86.6 Gt and 73.0 Gt for GSFC and CSR RL06 mascon solutions, respectively. This indicates that the noise reduction of the GCL-Mascon2024 solution is 35.5% and 23.5%, respectively, when compared to the GSFC and CSR RL06 mascon solutions. Rain shadow deserts are formed by the rain shadow effect. Orographic lift forces air masses to rise over mountains, cooling and losing moisture on the windward slopes. As the air

descends on the leeward side, it warms, increasing its moisture capacity and creating a drier region with reduced precipitation (Sun et al., 2008). The Taklamakan Desert, the largest in China, located in the rain shadow of the Himalayas, exemplifies this

phenomenon. The residuals of mass variations in this region are estimated to be 3.3 Gt, 2.1 Gt, 0.9 Gt, and 1.6 Gt, according to the GSFC RL06, CSR RL06, JPL RL06, and GCL-Mascon2024 mascon solutions, respectively. The Atacama Desert, a prime example of a coastal desert, is one of the driest regions on Earth, characterized by an almost complete absence of life

due to its extreme aridity. This hyperarid climate is primarily caused by the orographic effects of the Andes Mountains to the east and the Chilean Coast Range to the west, which prevent the desert from receiving significant precipitation. Additionally, the cold Humboldt Current and the persistent Pacific anticyclone play critical roles in maintaining the region's dryness. (Westbeld et al., 2009). The root mean square (RMS) of mass variations over the Atacama Desert, as derived from the mascon solutions by GSFC, CSR, and JPL, along with the GCL-Mascon2024, are 3.8 Gt, 1.8 Gt, 1.4 Gt, and 0.9 Gt, respectively,

indicating that the GCL-Mascon2024 solution has the smallest error.

Figure 11 illustrates the mass variations and the RMS of residuals of typical deserts. The deserts selected for this study include the Sahara Desert, Sechura Desert, Atacama Desert, Kyzylkum Desert, Gobi Desert, Taklamakan Desert, and Arabian Desert. The GCL-Mascon2024 incorporates well-defined physical constraints, such as coastal and basin boundaries, along with advanced spatial constraints based on the MVRCN matrix, enabling it to reduce errors in desert regions by approximately 28.1%

compared to the GSFC and CSR RL06 mascon solutions. Meanwhile, JPL RL06 mascon demonstrates slightly superior error suppression capability to the GCL-Mascon2024 solution in the aforementioned deserts. Notably, especially in the Atacama Desert, which is a long and narrow coastal desert from north to south and the driest desert in the world, GCL-Mascon2024 can achieve noise suppression ranging from 38.9% to 76.1% compared to the mascon solutions provided by GSFC, CSR, and JPL. As shown in Table 6, the noise SD of the mass-anomaly time series over the selected desert regions for the GSFC, CSR, JPL,

and GCL-Mascon2024 mascon solutions are 2.1 cm, 1.7 cm, 1.5 cm, and 1.1 cm, respectively. This translates to a random noise reduction of 86.1%, 50.8%, and 32.3% compared to the GSFC, CSR as well as JPL RL06 mascon solutions, respectively.

**Table 6. The root mean square of random noise over individual drainage systems of Greenland and desert regions from the mascon solutions recovered by GSFC, CSR, JPL, and GCL. The bolded value indicates the lowest RMS of random noise. (Unit: centimeters)**

| Region type | Drainage system / Basin name | GSFC | CSR | JPL | GCL |
|---|---|---|---|---|---|
| Polar region (Greenland) | North | **7.2514** | 8.7007 | 10.6156 | 7.7982 |
| | Northeast | 6.9647 | 6.8571 | 7.0732 | **5.8149** |
| | Northwest | **8.4266** | 19.0198 | 14.8654 | 10.8587 |
| | Central East | 7.3990 | 9.7565 | 7.2383 | **6.8310** |
| | Central West | **8.9223** | 12.6234 | 13.6435 | 10.0070 |
| | Southeast | 11.0628 | 19.0505 | 17.4096 | **9.1339** |
| | Southwest | **12.9805** | 20.5226 | 21.9191 | 14.9522 |
| Desert region | Sahara | 1.4999 | 1.1743 | 1.0664 | **0.7278** |
| | Arabian | 1.5333 | 1.5817 | 1.2517 | **0.7750** |
| | Gobi | 1.3932 | 0.9812 | 0.9095 | **0.7737** |
| | Kyzylkum | 2.4091 | 1.9919 | **1.5256** | 1.5724 |
| | Taklamakan | 1.6995 | 1.1447 | **0.5586** | 0.8951 |
| | Atacama | 2.9976 | 1.2463 | 1.3417 | **0.7556** |
| | Sechura | 2.9954 | 3.6538 | 3.6753 | **2.3070** |

## 5 Conclusions

Mascon solutions of Earth's temporal gravity field can be considered more "user-friendly" compared to spherical harmonic solutions, as they remove the need to apply empirical post-processing filters to eliminate errors in the unconstrained spherical harmonic solutions. Given this major advantage, mascon solutions have been garnering increased interest from the GRACE applications community. Herein, the Geodesy and Cryosphere Laboratory from the Southern University of Science and Technology presents a novel mascon solution named GCL-Mascon2024, derived utilizing the short-arc approach and the

Level-1B data from GRACE. The GCL-Mascon2024 features uniquely variable-shaped mascon geometries integrated with





relevant physical constraints such as coastline and basin boundary geometry, which ensures an accurate representation of temporal gravity signals while minimizing signal leakage. Meanwhile, this series of mascon recovery processes incorporate frequency-dependent data weighting techniques to reduce the influence of low-frequency noise in observations. GCL-Mascon2024 utilizes advanced spatial constraints based on the MVRCN matrix, which is constructed by integrating a priori

basin climate factors and cryosphere elevation models. The MVRCN matrix is carefully incorporated into the inversion process as a regularization matrix to minimize errors, ensuring the improvement of the signal-to-noise ratio in the GCL-Mascon2024 recovery framework.

To evaluate the quality of GCL-Mascon2024, we analyze the signal/error levels across continental regions globally, assess signal strengths over selected river basins and Greenland, and examine noise levels in representative desert areas. Based on

these analyses, we draw the following conclusions:

1.  Over the continental regions, GCL-Mascon2024 reduces the standard deviation of random errors by 6.5% to 20.4% compared to the RL06 mascon solutions provided by GSFC, CSR, and JPL. In particular, within non-humid river basins and desert regions, the GCL-Mascon2024 suppresses random noise by 36.7% and 56.4%, respectively, compared to contemporary mascon products.

2.  The mass changes and amplitudes from GCL-Mascon2024 over river basins and Greenland exhibit strong consistency with the RL06 mascon solutions from GSFC, CSR, and JPL, with correlation coefficients exceeding 95.0%, indicating good agreement in signal amplitudes across all four models. With SMB-based mass balance as the benchmark, GCL-Mascon2024 achieves a 10.1%−40.5% error reduction compared to the other three mascon solutions in the northern drainage system of Greenland, where mass variation is minimal.

3.  Mass variations in deserts, regions characterized by low precipitation, are typically minimal, offering an ideal basis for assessing the temporal errors of different mascon models. Building on this premise, the work investigates the error characteristics across diverse desert types, including the Sahara Desert (trade wind type), the Taklamakan Desert (rain shadow type), and the Atacama Desert (coastal type), along with other deserts. The GCL-Mascon2024 reduces temporal errors in these desert regions by approximately 28.1% compared to the RL06 mascon solutions from GSFC and CSR. In

particular, in the Atacama Desert—the world's driest and narrow coastal desert, extending from north to south—GCL-Mascon2024 achieves a noise reduction of 38.9% to 76.1% compared to the other three mascon solutions.

## 6 Data availability

The GRACE Level-1B data (downloaded from ftp://podaac.jpl.nasa.gov) and Kinematic orbits (available at ftp://ftp.tugraz.at) are given by JPL and Graz University of Technology, respectively. The ITSG-Grace2018 monthly solutions can be accessed

via: http://icgem.gfz-potsdam.de/series/03_other/ITSG/ITSG-Grace2018/monthly. The RL06 mascon solutions released by JPL, CSR, and GSFC are available respectively at https://podaac.jpl.nasa.gov/dataset/TELLUS_GRAC-GRFO_MASCON_ CRI_GRID_RL06.1_V3#, http://www2.csr.utexas.edu/grace/RL06_mascons.html, and https://earth.gsfc.nasa.gov/geo/data/ grace-mascons. The visualization tools for RL06 mascon products can be accessed through the following websites (https://ccar.colorado.edu/grace/about.html for JPL and GSFC RL06 mascon, last access: 25 September 2024; https://www2.

csr.utexas.edu/grace/RL06_Mascon_Viewer/Apps/index.php for CSR RL06 mascon, last access: 25 September 2024). The WaterGAP Global Hydrology Model for comparisons can be downloaded from https://doi.pangaea.de/10.1594/PANGAEA. 948461?format=html#download. The MAR (version 3.14) model used in this study can be downloaded from http://ftp.climato. be/fettweis/MARv3.14/Greenland/. The GCL-Mascon2024 model is available at https://doi.org/10.5281/zenodo.14008167 (Yan and Ran, 2024).



**Author contributions.**

Conceptualization: all; Formal analysis: ZY, JR, and PD; Funding acquisition: ZY and JR; Investigation: CS and XF; Methodology: ZY, JR, and PD; Software: ZY, JR, and PD; Supervision: JR, PD, and RK; Validation: PD, CS, RK, PS, and XF; Writing - original draft preparation: ZY, JR, and PD; Writing - review & editing: all.

**Competing interests.**

The contact author has declared that none of the authors have any competing interests.

**Disclaimer.**

Publisher's note: Copernicus Publications remains neutral with regard to jurisdictional claims made in the text, published maps, institutional affiliations, or any other geographical representation in this paper. While Copernicus Publications makes every effort to include appropriate place names, the final responsibility lies with the authors.

**Acknowledgements.**

We acknowledge the supercomputing resources supported by the Center for Computational Science and Engineering at the Southern University of Science and Technology.

**Financial support.**

This research has been supported by the National Natural Science Foundation of China (No. 42322403, 42174096, 41974094).



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
