# Peer review of "GCL-Mascon2024: a novel satellite gravimetry mascon solution using the short-arc approach"

_Earth System Science Data, 2024_

## Author Response (AR1)

**GCL-Mascon2024: a novel satellite gravimetry mascon solution using the short-arc approach**

Zhengwen Yan[1], Jiangjun Ran[1], Pavel Ditmar[2], C K Shum[3], Roland Klees[2], Patrick Smith[3], and Xavier Fettweis[4]

[1]Department of Earth and Space Sciences, Southern University of Science and Technology, Shenzhen, 518055, P. R. China

[2]Department of Geoscience and Remote Sensing, Delft University of Technology, Delft, 2628 CN, The Netherlands

[3]Division of Geodetic Science, School of Earth Sciences, The Ohio State University, Columbus, Ohio 43210, USA

[4]Department of Geography, University of Liège, Liège, B-4000, Belgium

submitted to *Earth System Science Data* (https://doi.org/10.5194/essd-2024-512)

**Responses to reviewers**

Dear Chief Editor, topic editor, reviewers, and community,

On behalf of all authors, we express our great appreciation to the Chief Editor, topic editor, reviewers, and community for their constructive and valuable comments and suggestions on our manuscript entitled "GCL-Mascon2024: a novel satellite gravimetry mascon solution using the short-arc approach" [ESSD-2024-512].

We have carefully studied the comments from reviewers and the community and then tried our best to revise our manuscript according to their valuable suggestions. The black text denotes the comments, while the red text contains our responses. Modifications made to the manuscript in response to these comments are highlighted in red italics. Besides, all the revised parts are in red in the revised paper. Please find the revised version attached, which we would like to submit for your kind consideration.

Hope you can consider a possible publication. We are looking forward to hearing from you. Thank you very much.

Yours sincerely,

Jiangjun Ran

**Anonymous Reviewer #1**

Summary

This paper presents a novel satellite gravimetry mascon solution named GCL-Mascon2024 for recovering the mass changes on the Earth's surface, which is the first to implement the short-arc approach for Mascon solution estimation. I commend the authors for their novel approach and encourage them to continue refining and expanding upon this exciting methodology. The research findings are highly innovative and scientifically valuable and are of great significance for the research of the Earth's gravity field and the development of related fields. The paper has a complete structure, clear logic, reasonable experimental design, and detailed data, providing new ideas and methods for follow-up research. However, there is still room for improvement. I would like to recommend minor revisions of the manuscript before publication in Earth System Science Data, according to the comments as follows.

**Response:**

Thank you very much for your constructive comments on our manuscript. There is no doubt that these comments are valuable and very helpful for revising and improving our manuscript. We carefully modified the manuscript based on your comments and suggestions. Please kindly refer to the following text for more details. We are deeply grateful for your recognition and support of our work.

Comments

I would like to know whether the gradient correction, a well-established component of the classical short-arc approach, was incorporated into the Mascon solution process. If gradient correction was applied, I recommend the authors provide a detailed description of the strategy used. If gradient correction was not applied, the authors should justify this decision. This additional information would enhance the methodological transparency and allow readers to better evaluate the robustness of the proposed approach.

**Response:**

We sincerely appreciate your constructive comments. We have modified the main text

to clarify the use of the gradient correction algorithm. Please kindly refer to the following text (Lines 276-284 in the revised manuscript).

*To determine the appropriate arc length for GCL-Mascon2024, we conducted computations of a monthly mascon model using different arc lengths to compare the stability of the resulting estimates. Table 2 presents the condition numbers of the unconstrained normal matrices and the corresponding computational time needed for different arc lengths. From this standpoint, the 2-hr arc length corresponds to the most stable arc length in the GCL-Mascon2024 recovery. Figure 4 illustrates that increasing the arc length beyond 2 hr in the short-arc approach leads to a significant increase in noise in gravity field estimates as the normal equations become more ill-conditioned. This observation aligns closely with what we conclude from Table 2. Therefore, an arc length of 2-hr is determined to be the most suitable for the short-arc approach employed in this work. Additionally, we incorporate the gradient correction algorithm proposed by Mayer-Gürr (2008) to consider the influence of the kinematic orbit errors.*

Reference

Mayer-Gürr, T.: Gravitationsfeldbestimmung aus der Analyse kurzer Bahnbögen am Beispiel der Satellitenmissionen CHAMP und GRACE, Rheinische Friedrich-Wilhelms-Universität Bonn, Landwirtschaftliche Fakultät, IGG-Institut für Geodäsie und Geoinformation, 2008.

The noise of GCL-Mascon2024 in the Caspian Sea and northern Australia is very low. This is an intriguing result that warrants further investigation. It is advisable to conduct an in-depth analysis centering around the Caspian Sea, highlighting the performance of your solution and improving the analysis.

**Response:**

We sincerely appreciate your valuable comments. Following your suggestion, we have explained an in-depth analysis centering around the Caspian Sea in the revised manuscript (Lines 496-509). Please kindly refer to the following text for more details.

*The utilization of mass variations in large lakes (e.g., the Caspian Sea) to assess noise levels in GRACE solutions is a well-established approach (e.g., Loomis and Luthcke, 2017; Ditmar, 2022). Herein, we choose the largest lake on Earth, the Caspian Sea, as*

*an example for verification. We follow the approach proposed by Ditmar (2022), wherein the mass anomaly time series derived from GRACE is compared with the water level time series obtained from satellite altimetry observations. The latter time series is empirically rescaled (with a scaling factor of 0.687 for the Caspian Sea provided by Ditmar (2022)) to account for signal damping in the GRACE solution. Figure 12 presents the mass anomaly time series over the Caspian Sea derived from various mascon solutions and satellite altimetry data. As illustrated, the GCL-Mascon2024 solution shows strong consistency with the other models in capturing mass variations in this region. Using satellite altimetry-derived mass variations, scaled by a factor of 0.687, as the reference, the noise SD for the GSFC, CSR, JPL, and GCL-Mascon2024 mascon solutions are 5.7 cm, 5.8 cm, 5.6 cm, and 5.2 cm, respectively.*

[Figure]

*Figure 12. Comparison of GRACE-derived mass anomaly time series (expressed in equivalent water height, EWH) from different mascon solutions with satellite altimetry-based water level variations over the Caspian Sea. The time series derived from satellite altimetry has been downscaled using a scale factor of 0.687 to account for signal attenuation (Ditmar, 2022).*

It is recommended to add comparisons of the residuals in the open ocean.

**Response:**

We sincerely appreciate your valuable comments. Following your suggestion, we have supplemented the comparison in the time series of ocean mass. Please kindly refer to the following text (Lines 510-517 in the revised manuscript).

*GRACE satellite gravity measurements over oceanic regions directly correspond to ocean bottom pressure variations at spatial scales of ~300 km (Watkins et al., 2015).*

*Figure 13 illustrates the time series of basin mass variations derived from different mascon solutions. To assess the quality of our solutions for ocean signals, we compute the correlation coefficients between GCL-Mascon2024 and the RL06 mascon solutions released by GSFC, CSR, and JPL. The resulting correlations are 95.7%, 98.0%, and 98.2%, respectively, indicating a high level of consistency between our products and official mascon products.*

[Figure]

*Figure 13. Comparison of GRACE-derived mass anomaly time series (expressed in equivalent water height, EWH) over the global sea from different mascon solutions.*

Reference

Watkins, M. M., Wiese, D. N., Yuan, D.-N., Boening, C., and Landerer, F. W.: Improved methods for observing Earth's time variable mass distribution with GRACE using spherical cap mascons, Journal of Geophysical Research-Solid Earth, 120, 2648-2671, https://doi.org/10.1002/2014jb011547, 2015.

Please clarify the time interval used for constructing the observation equation, particularly in light of the differing sampling rates between the kinematic orbit (10 seconds) and other L1B data (5 seconds). Specifically, address how these discrepancies in sampling rates are reconciled.

**Response:**

We sincerely appreciate your valuable comments. As your comments pointed out, the kinematic orbit sample rate is different from the other L1B data. The integration time interval is 5 seconds for the observation equation using the rangerate observation, while

the integration time interval of the observation equation based on the orbit is the same as the kinematic orbit sample rate. The purpose of this strategy is to include as much rangerate data as possible into the GCL-Mascon2024 temporal gravity field determination.

Kindly provide a detailed explanation of the error assessment strategy employed for the kinematic orbits. This should include the following: -The criteria used to identify and classify errors, -Whether interpolated epochs are incorporated into constructing the observation equation, etc.

**Response:**

We sincerely appreciate your valuable comments. It is well-established that kinematic orbits contain a lot of gross errors. Firstly, we use the Pauta criterion to give a quality flag to the kinematic orbit. The gross error data quality mark is 0, and the normal data quality mark is 2. Secondly, as for the gap in the kinematic orbits, we fill the gap with the reduced dynamic orbit (i.e., GNV1B data), and its quality mark is 1. Lastly, different weights are assigned to orbit data with different quality tags to construct the observation equation: 2 corresponds to the maximum weight, 1 to an intermediate weight, and 0 to the minimum weight.

Minor comments

Page 3, Lines 93-95: Section 5 is information on the dataset, and section 6 is the conclusion. These two parts are reversed in the text. Please adjust the order to ensure consistency.

**Response:**

We sincerely appreciate your valuable comments. We have adjusted the order of sections 5 and 6 in the revised version of the manuscript (Lines 99-101 in the revised manuscript).

Page5, Line 157, Table 1: Please explain the similarities and differences between the background force model in the Mascon solution and the spherical harmonic solution.

**Response:**

We sincerely appreciate your valuable comments. Both the Mascon and spherical harmonic solutions utilize identical background force models during the processing of Level-1B data. These shared models include the solid (pole) Earth and ocean (pole) tides, nontidal atmosphere and ocean dealiasing, Atmospheric tides, third-body attractions, and general relativity. The consistency in these foundational models ensures that both approaches adhere to the same standards for gravity field recovery. While the core background force models remain aligned, the Mascon solution requires five additional corrections to account for specific geophysical and geometric effects not inherently resolved by the unfiltered spherical harmonic approach:

(1) Earth's Elastic Response

Mascon solutions explicitly incorporate the elastic response of the solid Earth to surface mass redistribution. This correction accounts for instantaneous crustal deformation induced by surface loading, which is critical for isolating true mass signals from geometric displacements.

(2) Glacial Isostatic Adjustments

GIA correction is applied to mitigate the viscoelastic rebound of the Earth's mantle due to Pleistocene deglaciation. This long-term signal, often conflated with contemporary mass changes in gravity solutions, is explicitly modeled and removed in Mascon recovery.

(3) Earth Ellipsoidal Corrections

The Earth's oblate shape necessitates ellipsoidal corrections to accurately represent mass anomalies on the reference ellipsoid rather than a spherical surface. These geometric adjustments ensure consistency with the Earth's true gravitational potential field.

(4) Low-degree Term Corrections

Mascon solutions apply targeted corrections to low-degree spherical harmonic terms (e.g., degree-1 and degree-2 coefficients) to address systematic errors arising from satellite orbit parameterization and reference frame uncertainties.

(5) GAD Corrections

To explicitly contain seafloor pressure anomalies in the corrected mascon solutions, the AOD1B RL06 GAD product (Dobslaw et al., 2017) is reintegrated into the

mascon calibration framework.

Following a standardized processing workflow (Watkins et al., 2015; Save et al., 2016; Loomis et al., 2019; Tregoning et al., 2022), the uncorrected mascon solutions (i.e., $\text{MASCON}_{Uncorrected}$, we will return to that point in Sect. 2.5) are systematically integrated with the aforementioned corrected components to generate corrected mascon grids. The formula to generate the corrected mascon grid is

$$\text{MASCON}_{Corrected} = \text{MASCON}_{Uncorrected} - \text{MASCON}_{C_{20}} + \text{SLR}_{C_{20}} + \text{DEG1} - \text{GIA} + \text{GAD} . \quad (4)$$

Reference

Dobslaw, H., Bergmann-Wolf, I., Dill, R., Poropat, L., Thomas, M., Dahle, C., Esselborn, S., Koenig, R., and Flechtner, F.: A new high-resolution model of non-tidal atmosphere and ocean mass variability for de-aliasing of satellite gravity observations: AOD1B RL06, Geophysical Journal International, 211, 263-269, https://doi.org/10.1093/gji/ggx302, 2017.

Loomis, B. D., Luthcke, S. B., and Sabaka, T. J.: Regularization and error characterization of GRACE mascons, Journal of Geodesy, 93, 1381-1398, https://doi.org/10.1007/s00190-019-01252-y, 2019.

Save, H., Bettadpur, S., and Tapley, B. D.: High-resolution CSR GRACE RL05 mascons, Journal of Geophysical Research-Solid Earth, 121, 7547-7569, https://doi.org/10.1002/2016jb013007, 2016.

Tregoning, P., McGirr, R., Pfeffer, J., Purcell, A., McQueen, H., Allgeyer, S., and McClusky, S. C.: ANU GRACE Data Analysis: Characteristics and Benefits of Using Irregularly Shaped Mascons, Journal of Geophysical Research-Solid Earth, 127, https://doi.org/10.1029/2021jb022412, 2022.

Watkins, M. M., Wiese, D. N., Yuan, D.-N., Boening, C., and Landerer, F. W.: Improved methods for observing Earth's time variable mass distribution with GRACE using spherical cap mascons, Journal of Geophysical Research-Solid Earth, 120, 2648-2671, https://doi.org/10.1002/2014jb011547, 2015.

Page12, Line 320: annual amplitude -> annual amplitudes

**Response:**

We sincerely appreciate your valuable comments. We have corrected this typo in the revised version of the manuscript (Line 338 in the revised manuscript).

Page 16, Table 5: The table currently presents with four decimal places. However, such

precision does not appear necessary for this study's context. To improve clarity and readability, it is recommended to round the values to 1 decimal place or, at most, 2 decimal places.

**Response:**

We sincerely appreciate your valuable comments. As recommended, we have revised Table 5 by rounding all numerical values to two decimal places. Please kindly refer to the revised manuscript (Line 405) for more details.

Page 19, Table6: Similar to the above comment. Please revise the table 6 accordingly.

**Response:**

We sincerely appreciate your valuable comments. Following your suggestion, we have revised Table 6 by rounding all numerical values to two decimal places. Please kindly refer to the revised manuscript (Line 481) for more details.

Page 18, section 4.2.3: Please explain why the climate component needs to be removed from the desert for the Mascon solutions validation and assessment and add the necessary references.

**Response:**

We sincerely appreciate your valuable comments. We have supplemented the reasons why the climate component needs to be removed from the desert. Please kindly refer to the following text (Lines 448-452 in the revised manuscript) for more details.

*Our impetus stems from an understanding that precipitation within desert regions is limited. It is critical to emphasize that aridity cannot be equated with negligible temporal mass variations (e.g., Scanlon et al., 2022). Conversely, low precipitation may stimulate an extensive consumption of groundwater. To that end, the residuals, calculated after removing the climatological components (i.e., bias, trend, and amplitude) from the mass variations, can be regarded as mis-modeling signals or temporal noise that persist in the temporal gravity fields (e.g., Zhou et al., 2024).*

Reference
Scanlon, B. R., Rateb, A., Anyamba, A., Kebede, S., MacDonald, A. M., Shamsudduha, M., Small, J., Sun, A., Taylor, R. G., and Xie, H.: Linkages between GRACE water storage,

hydrologic extremes, and climate teleconnections in major African aquifers, Environmental Research Letters, 17, https://doi.org/10.1088/1748-9326/ac3bfc, 2022.

Zhou, H., Zheng, L., Li, Y., Guo, X., Zhou, Z., and Luo, Z.: HUST-Grace2024: a new GRACE-only gravity field time series based on more than 20 years of satellite geodesy data and a hybrid processing chain, Earth System Science Data, 16, 3261-3281, https://doi.org/10.5194/essd-16-3261-2024, 2024.

Page 20, section 6: I recommend including the access dates for all datasets, which ensures readers and future researchers can trace the exact versions of the data used in the study.

**Response:**

We sincerely appreciate your valuable comments. Following your suggestion, we have supplemented the access dates for all datasets. Please kindly refer to the following text (Line 551 in the revised manuscript).

*All datasets used in this study were last accessed on 25 May 2025. The specific data repositories include: …*

P25, Lines 794-796: The manuscript generally maintains a high referencing standard; however, I noticed inconsistencies in the formatting of author names in the reference list. e.g., in Line 794, the author is cited as "Wiese, D.," while in Line 796, the same author is cited as "Wiese, D. N.". I recommend carefully reviewing the entire reference list and standardizing the formatting of author names.

**Response:**

We sincerely appreciate your valuable comments. Following your suggestion, we have thoroughly reviewed the entire reference list and corrected all inconsistencies in the formatting of author names. Please kindly refer to the revised manuscript for more details.

**Anonymous Reviewer #2**

The manuscript provides a new approach to estimate a mascon-based GRACE product. The manuscript is well organized and the technical details are clearly presented. It has a high potential to contribute to this field and to raise widespread interest. However, the results have some unique features that deviate from the previous mascon products and I am curious about their reliability.

**Response:**

We wish to express our sincere gratitude for your thorough review and insightful comments on our manuscript. Your professional feedback has provided crucial guidance for enhancing the academic rigor and quality of our work. We have diligently incorporated all suggested revisions and all modifications clearly highlighted in the manuscript's track-changes version. We appreciate your earnest work and hope that the corrections will meet with approval. Once again, thank you very much for your warm work and constructive comments.

1) the seemingly high spatial resolution of the product in this study (GCL-mascon2024) apparently comes from the regularization constraint shown in Fig. 3. This strong and time-invariant a priori information could lead to several problems. First, the distribution of the regularization agrees well with the seasonality in mass changes, but deviates from the long-term trends, implying that the trend will be less constrained. It is strange that the authors didn't provide a comparison of the trend maps. Second, the oceans are not constrained, and the authors didn't provide the results in oceans either. I suspect that the results in oceans may not be as good as on land. Third, I don't get the rationale for placing a topographic constraint on ice sheets.

**Response:**

We sincerely appreciate your constructive comments. "This strong and time-invariant a priori information could lead to several problems.", as astutely noted by the reviewer, presents a direction of future research for our mascon recovery from GRACE/GRACE-FO observations. While our current mascon recovery framework, employing time-invariant constraints, demonstrates the capability to effectively resolve temporal gravity field signals, we fully acknowledge the scientific merit of incorporating time-varying

regularization constraints as a direction for future methodological refinement. Our team is currently developing a dedicated experimental protocol to systematically explore the potential advantages of time-varying signal recovery under varying time-variant information constraints.

**Regarding your first comment**, "First, the distribution of the regularization agrees well with the seasonality in mass changes, but deviates from the long-term trends, implying that the trend will be less constrained. It is strange that the authors didn't provide a comparison of the trend maps.", we have supplemented a comparison of the trend maps. Please kindly refer to the following text (Lines 334-337 in the revised manuscript).

*In Figure 6, we specifically present the long-term trends in temporal gravity signals for the time span ranging from January 2003 to December 2015. Upon observing Figure 6, we can discern a high level of consistency in the global mass change signals across all four models.*

[Figure]

*Figure 6. Long-term trends from January 2003 to December 2015 (in equivalent water height, or EWH).*

**Regarding your second comment**, "Second, the oceans are not constrained, and the authors didn't provide the results in oceans either. I suspect that the results in oceans may not be as good as on land.", we have explained how the values in oceans are

derived in the revised manuscript (Lines 158-162 of the revised manuscript) and the analysis of the ocean signals (Lines 510-517 of the revised manuscript).

*Following a standardized processing workflow (Watkins et al., 2015; Save et al., 2016; Loomis et al., 2019; Tregoning et al., 2022), the uncorrected mascon solutions (i.e., $MASCON_{Uncorrected}$, we will return to that point in Sect. 2.5) are systematically integrated with the aforementioned corrected components to generate corrected mascon grids. The formula to generate the corrected mascon grid is*

$$MASCON_{Corrected} = MASCON_{Uncorrected} - MASCON_{C_{20}} + SLR_{C_{20}} + DEG1 - GIA + GAD. \quad (4)$$

*GRACE satellite gravity measurements over oceanic regions directly correspond to ocean bottom pressure variations at spatial scales of ~300 km (Watkins et al., 2015). Figure 13 illustrates the time series of basin mass variations derived from different mascon solutions. To assess the quality of our solutions for ocean signals, we compute the correlation coefficients between GCL-Mascon2024 and the RL06 mascon solutions released by GSFC, CSR, and JPL. The resulting correlations are 95.7%, 98.0%, and 98.2%, respectively, indicating a high level of consistency between our products and official mascon products.*

[Figure]

*Figure 13. Comparison of GRACE-derived mass anomaly time series (expressed in equivalent water height, EWH) over the global sea from different mascon solutions.*

**Regarding your third comment**, "Third, I don't get the rationale for placing a topographic constraint on ice sheets.", we would like to use the Greenland Ice Sheet (GIS) as an example to explain why the constraint matrix is designed based on the

topography of the ice sheet. The GIS mass evolution is dominated by the signal from the coastal margins of the ice sheet (Luthcke et al., 2013). Below 2000 m elevation, significant coastal mass loss is primarily driven by ice discharge from fast-flowing, marine-terminating outlet glaciers in the northwest, southwest, and southeastern regions of the Greenland Ice Sheet (Moon et al., 2012); above this elevation threshold, no statistically significant mass variations are observed (Luthcke et al., 2013). In the GSFC mascon solution (Luthcke et al., 2013; Loomis et al., 2019), the Greenland Ice Sheet is partitioned into two constraint regions based on elevation thresholds: (1) areas below 2000 m and (2) areas above 2000 m. Figure R1 depicts the Greenland Ice Sheet topography (Figure R1-a) and the mascon-scale regularization matrix (Figure R1-b) used in the GCL-Mascon2024 recovery framework. A notable spatial alignment is observed between regions with stronger regularization constraints (dark blue areas in Figure R1-b) and the 2000 m elevation contour zone of the ice sheet.

[Figure]

Figure R1. Comparison of Greenland Ice Sheet topography and regularization constraint matrix: (a) topographic map, (b) regularization constraint matrix

Reference

Loomis, B. D., Luthcke, S. B., and Sabaka, T. J.: Regularization and error characterization of GRACE mascons, Journal of Geodesy, 93, 1381-1398, https://doi.org/10.1007/s00190-019-01252-y, 2019.

Luthcke, S. B., Sabaka, T. J., Loomis, B. D., Arendt, A. A., McCarthy, J. J., and Camp, J.:

Antarctica, Greenland and Gulf of Alaska land-ice evolution from an iterated GRACE global mascon solution, Journal of Glaciology, 59, 613-631, https://doi.org/10.3189/2013JoG12J147, 2013.

Moon, T., Joughin, I., Smith, B., and Howat, I.: 21st-Century Evolution of Greenland Outlet Glacier Velocities, Science, 336, 576-578, https://doi.org/10.1126/science.1219985, 2012.

2) The results of this study seem to be always smaller than the other mascons (Fig. 9, 10). I feel a stronger regularization is imposed in this study. How about the series of total mass changes in Greenland and Antarctica?

**Response:**

We sincerely appreciate your constructive comments. Figure R2 and Figure R3 show the time series of total mass change in Greenland and Antarctica. Please kindly refer to the following figures for more details.

[Figure]

Figure R2. Time series of total mass changes in Greenland

[Figure]

Figure R3. Time series of total mass changes in Antarctica

3) The comparison in Apr. 2008 (Fig. 6) looks like GCL-mascon2024 could provide

more details, .e.g, in the tropical Africa, Bangladesh and Amazon. But some signals seem to be removed, e.g., in Caspian and Madagascar. Please add more regional comparisons and use hydrological models to evaluate the differences. Comparison for more epochs is also recommended.

**Response:**

We sincerely appreciate your constructive comments. Following your suggestion, we have supplemented more regional comparisons and used hydrological models to evaluate the differences. Besides, comparisons for more epochs are also provided. At the same time, we also extended the mascon solution period from December 2012 to December 2015. Following your suggestion, we have explained an in-depth analysis centering around the Caspian Sea in the revised manuscript (Lines 496-509). Please kindly refer to the following figures (i.e., Figure R4~Figure R6, Figure 12) for more details.

[Figure]

Figure R4. Time series of mass anomalies over typical river basins from the hydrology model WaterGAP and mascon solutions recovered by GSFC, CSR, JPL, and GCL

[Figure]

Figure R5. Global mass change signals during April 2008 from the hydrology model WaterGAP and mascon solutions recovered by GSFC, CSR, JPL, and GCL

[Figure]

Figure R6. Global mass change signals during May 2009 from the hydrology model WaterGAP and mascon solutions recovered by GSFC, CSR, JPL, and GCL

*The utilization of mass variations in large lakes (e.g., the Caspian Sea) to assess noise levels in GRACE solutions is a well-established approach (e.g., Loomis and Luthcke, 2017; Ditmar, 2022). Herein, we choose the largest lake on Earth, the Caspian Sea, as an example for verification. We follow the approach proposed by Ditmar (2022), wherein the mass anomaly time series derived from GRACE is compared with the water level time series obtained from satellite altimetry observations. The latter time series is empirically rescaled (with a scaling factor of 0.687 for the Caspian Sea provided by Ditmar (2022)) to account for signal damping in the GRACE solution. Figure 12 presents the mass anomaly time series over the Caspian Sea derived from various mascon solutions and satellite altimetry data. As illustrated, the GCL-Mascon2024 solution shows strong consistency with the other models in capturing mass variations in this region. Using satellite altimetry-derived mass variations, scaled by a factor of 0.687, as the reference, the noise SD for the GSFC, CSR, JPL, and GCL-Mascon2024 mascon solutions are 5.7 cm, 5.8 cm, 5.6 cm, and 5.2 cm, respectively.*

[Figure]

*Figure 12. Comparison of GRACE-derived mass anomaly time series (expressed in equivalent water height, EWH) from different mascon solutions with satellite altimetry-based water level variations over the Caspian Sea. The time series derived from satellite altimetry has been downscaled using a scale factor of 0.687 to account for signal attenuation (Ditmar, 2022).*

The comparison in Annual amplitude (Fig. 7) also raises some interesting phenomenon. Again, GCL-mascon2024 shows are details, but also numerous speckles, in the south of the Sahara Desert, in the U.S, and in south China. Some signals are missing, like, in California, Caspian (I feel the authors directly attribute zero to this region), areas

surrounding the Gulf of Carpentaria. Some signals are newly revealed, like, in eastern Europe.

**Response:**

We sincerely appreciate your constructive comments. Following this comment, we have extended the temporal coverage through Dec. 2015 and subsequently recomputed the annual amplitude for the four mascon solutions, with the updated results presented in the following figure.

[Figure]

*Figure 7. Annual amplitudes from January 2003 to December 2015
(in equivalent water height, or EWH).*

4) there should be comparison in the time series of ocean mass and a check on the mass conservation at the global scale.

**Response:**

We sincerely appreciate your constructive comments. Following your suggestion, we have supplemented the comparison in the time series of ocean mass. Please kindly refer to the following text (Lines 510-517 in the revised manuscript).

*GRACE satellite gravity measurements over oceanic regions directly correspond to ocean bottom pressure variations at spatial scales of ~300 km (Watkins et al., 2015). Figure 13 illustrates the time series of basin mass variations derived from different mascon solutions. To assess the quality of our solutions for ocean signals, we compute the correlation coefficients between GCL-Mascon2024 and the RL06 mascon solutions*

*released by GSFC, CSR, and JPL. The resulting correlations are 95.7%, 98.0%, and*

*98.2%, respectively, indicating a high level of consistency between our products and*

*official mascon products.*

[Figure]

*Figure 13. Comparison of GRACE-derived mass anomaly time series (expressed in equivalent water height, EWH) over the global sea from different mascon solutions.*

In accordance with this recommendation, we have conducted rigorous validation analyses on a global scale. Figure R7 shows the time series of global mass variations, demonstrating exceptional consistency between the GCL-Mascon2024 solution and established JPL/CSR mascon products, with correlation coefficients reaching 99.1% (vs. JPL mascon) and 98.9% (vs. CSR mascon). The high level of consistency between GCL-Mascon2024 and the official mascon product (e.g., JPL and CSR mascon) confirms the mass conservation of our solution on a global scale.

[Figure]

Figure R7. Comparison of GRACE-derived mass anomaly time series (expressed in equivalent water height, EWH) at the global scale from different mascon solutions.

Reference

Watkins, M. M., Wiese, D. N., Yuan, D.-N., Boening, C., and Landerer, F. W.: Improved methods for observing Earth's time variable mass distribution with GRACE using spherical cap mascons, Journal of Geophysical Research-Solid Earth, 120, 2648-2671, https://doi.org/10.1002/2014jb011547, 2015.

5) The figures are of poor quality. The maps are too small and the time series are blurred. This seriously affects the correct assessment of the quality of the results.

**Response:**

We sincerely appreciate your constructive comments. In response to your critical observation regarding suboptimal figures' clarity, we have rigorously enhanced the quality of all cartographic representations and elevated the image resolution to 600 dpi in accordance with the "Earth System Science Data" visualization standards. Please kindly refer to the revised manuscript for more details.

--- other comments

L76, it should be explained the potential advantage of using short-arc.

**Response:**

We sincerely appreciate your valuable comments. We have supplemented the potential advantage of using short-arc, please kindly refer to the following text (Lines 78-83 in the revised manuscript).

*This study represents the first application of the short-arc approach to recover the global mascon solution. A distinguishing feature of this methodology compared to other conventional approaches lies in its substantially reduced arc length integration interval (Mayer-Gürr et al., 2005; Mayer-Gürr, 2008). The temporal gravity field based on the short-arc approach exhibits enhanced stability and superior accuracy owing to the substantially reduced condition number of the normal equation system (Chen et al., 2015).*

Reference

Chen, Q., Shen, Y., Zhang, X., Hsu, H., Chen, W., Ju, X., and Lou, L.: Monthly gravity field models derived from GRACE Level 1B data using a modified short-arc approach,

Journal of Geophysical Research-Solid Earth, 120, 1804-1819, https://doi.org/10.1002/2014jb011470, 2015.

Mayer-Gürr, T.: Gravitationsfeldbestimmung aus der Analyse kurzer Bahnbögen am Beispiel der Satellitenmissionen CHAMP und GRACE, Rheinische Friedrich-Wilhelms-Universität Bonn, Landwirtschaftliche Fakultät, IGG-Institut für Geodäsie und Geoinformation, 2008.

Mayer-Gürr, T., Ilk, K. H., Eicker, A., and Feuchtinger, M.: ITG-CHAMP01:: a CHAMP gravity field model from short kinematic arcs over a one-year observation period, Journal of Geodesy, 78, 462-480, https://doi.org/10.1007/s00190-004-0413-2, 2005.

Equ(1) rho_i is not explained

**Response:**

We sincerely appreciate your valuable comments. We have supplemented the explanation of rho_i, please kindly refer to the following text (Lines 116-117 of the revised manuscript).

*The temporal gravity field can be modeled as a series of N mascons, with the surface mass density (mass per unit area) of mascon $M_i$ represented by $\rho_i$ (i=1, 2, ..., N).*

L130, I could not see basin boundaries play a role in Fig. 1.

**Response:**

We sincerely appreciate your valuable comments. We have provided a schematic diagram (Figure R8) that contrasts the mascon partitioning schemes with and without basin boundary constraints. Figure R8 explicitly visualizes this distinction:

Figure R8-a: This scheme defines coastlines as the sole natural boundaries for mascon delineation, while mascon partitioning remains homogeneous across continental interiors.

Figure R8-b: Coastlines and basin boundaries serve as critical spatial constraints to align mascons with natural hydrological domains, thereby minimizing signal leakage across basins.

[Figure]

Figure R8. Schematic illustration of different Mascon partitioning.
(a) coastlines constraint; (b) coastlines and basin boundaries constraint

L133, "no signal correlation between basins", I am curious to see whether it is physically true. Have the authors carried out numerical experiments based on models to confirm this? Besides, where did the authors apply this non-correlation criterion in the inversion?

**Response:**

We sincerely appreciate your valuable comments. In response to your first comment regarding "Have the authors carried out numerical experiments based on models to confirm this?", we would like to respond to this comment from the perspective of parameterization and regularization.

**1. Simulation experiment based on the parameterized scheme of "no signal interaction between river basins"**

Based on the two parameterization schemes shown in Figure R8 (i.e., coastline constraint parameterization vs. coastlines and basin boundaries constraint parameterization), we design simulation experiments to verify the potential improvement brought by the basin boundaries constraint parameterization. The data used in the simulation are shown in Table R1. Herein, we use the Greenland Ice Sheet as an example to show the impact of different parameterizations on solution accuracy.

As shown in Figure R9, compared with the mascon with coastlines constraint, the mascon with coastlines and basin boundaries constraint can reduce the error by 25.2% in the long-term trend of the temporal signal.

Table R1. Summary of Signals and Errors in the Simulation Experiment

| Data Types | Characteristics |
|---|---|
| *Signals* | |
| Greenland ice sheet and Antarctic ice sheet | Combination of detrend RACMO2.3p2 and long-term trend derived from altimetry data |
| Other continent regions | PCR-GLOBWB |
| Regional glaciers and lakes | Altimetry data |
| Ocean | Principle of global mass conservation |
| *Errors* | |
| Aliasing noise | AOD noise |
| Parameterization error | True signals and inversion solutions with different spatial resolution |

[Figure]

Figure R9. Long-term trend error from different Mascon partitioning.
(a) coastlines constraint; (b) coastlines and basin boundaries constraint

Reference

Dong Fang, Jiangjun Ran, Shin-Chan Han, Natt Tangdamrongsub, Zhengwen Yan. An optimized parameterized Mascon solution for estimating surface mass changes from GRACE(-FO) satellite gravimetry, Geophysical Journal International (Under review)

**2. Simulation experiment based on the regularization scheme of "no signal interaction between river basins"**

The detailed configuration parameters of this numerical simulation are comprehensively documented in our research team's peer-reviewed publication "Analysis and mitigation of biases in Greenland ice sheet mass balance trend estimates from GRACE mascon products" (Ran et al., 2021) in the Journal of Geophysical Research: Solid Earth.

The "true" trend signal for the NW drainage system is shown in Figure R10-a, whereas the estimates obtained with the ordinary and basin-boundary constraint spatial constraints and two different regularization parameters (i.e., $10^{-24}$ and $10^{-25}$) are plotted in Figure R10 b-e. In addition, the differences between the estimates and the true trends are shown in Figure R11. We see that the biases when using ordinary spatial constraints are significantly larger than those when using the basin-boundary constraint spatial constraints, especially when the regularization parameter increases from $10^{-25}$ to $10^{-24}$. When the regularization parameter is set equal to $10^{-25}$, the RMS of the residuals between the estimated and the true signal per drainage system decreases by a factor of 15% (namely, from 1.3 to 1.1 Gt/yr) after switching from the ordinary spatial constraints to the basin-boundary constraint ones. The reduction becomes even larger when the regularization parameter is increased to $10^{-24}$ (namely, from 6.4 to 2.8 Gt/yr, i.e., by 56%). The results for the other drainage systems are similar, please kindly refer to Table R2 for more details.

Table R2. RMS differences between the estimated and true trends-per-mascon, reflecting the biases caused by the ordinary and basin-boundary constraint spatial regularization

| | | N | NW | NE | SW | SE |
|---|---|---|---|---|---|---|
| Ordinary | RMS for $\alpha = 10^{-25}$ | 0.4 | 1.3 | 1.9 | 0.9 | 2.3 |
| regularization | RMS for $\alpha = 10^{-24}$ | 1.5 | 6.4 | 3.3 | 3.1 | 6.6 |
| Basin-boundary constraint | RMS for $\alpha = 10^{-25}$ | 0.3 | 1.1 | 1.7 | 0.8 | 2.3 |
| regularization | RMS for $\alpha = 10^{-24}$ | 1.2 | 2.8 | 4.2 | 2.4 | 3.7 |
| Bias | for $\alpha = 10^{-25}$ | 25% | 15% | 11% | 11% | 0% |
| reduction | for $\alpha = 10^{-24}$ | 20% | 56% | -27% | 23% | 44% |

PS: The RMS differences are computed per drainage system. In each case, the true signal is limited to only one drainage system, as indicated by the name of the corresponding column. The first and

second rows show the RMS biases after the ordinary spatial regularization for the two regularization parameters under consideration. The third and fourth rows present similar information in the case of basin-boundary constraint spatial regularization. The fifth and sixth rows refer to the reduction of these RMS differences when the basin-boundary constraint regularization is compared with the ordinary one (a negative number corresponds to an increase in the RMS difference). The unit is Gt/yr. (Ran et al., 2021)

[Figure]

Figure R10. True (panel a) versus estimated trends (panels b–e) for ordinary and basin-boundary constraint first-order Tikhonov regularization with two regularization parameters (ordinary regularization vs. basin-boundary constraint regularization) (Ran et al., 2021)

[Figure]

Regularization parameter $10^{-24}$

(a) ordinary regularization

(b) improved regularization

Regularization parameter $10^{-25}$

(c) ordinary regularization

(d) improved regularization

Figure R11. Trend signal of spectral leakage from the NW drainage system for ordinary and first-order Tikhonov regularization with two regularization parameters (Ran et al., 2021)

Actually, the ANU mascon released by the Australian National University (ANU) attributes part of its accuracy advantage to the variable-shaped mascon geometry that does not cross coastlines or Amazon basin boundaries (Allgeyer et al., 2022; Tregoning et al, 2022).

In response to your second comment regarding "Besides, where did the authors apply this non-correlation criterion in the inversion?", we would like to present the following responses.

In the GCL-Mascon2024 recovery framework, the non-correlation in different drainage systems criterion is introduced as a parameterization in the inversion. As described in Section 2.3, the surface area (i.e., $S_i$) and the number of the Fibonacci nodes (i.e., $K_i$) of mascon $M_i$ in the inversion formula (2) are important parameters determined by the aftermentioned criterion. The expression of formula (2) is

$$\hat{I}_{i,p} \approx \sum_{j=1}^{K_i} \frac{S_i}{K_i \cdot \left( l_{ij,p} \right)^2} \cdot \hat{d}_{ij,p} \,. \tag{2}$$

where $l_{ij,p}$ represents the distance between a Fibonacci point $j$ located in the mascon $M_i$ and the satellite measurement point $p$; $\hat{d}_{ij,p}$ is a unit vector pointing from the satellite measurement point $p$ to a Fibonacci point $j$ located in the mascon $M_i$. One important thing we would like to clarify is that Fibonacci points are integration points of the Newton-Cotes formula (Gonzalez, 2010).

Reference

Allgeyer, S., Tregoning, P., McQueen, H., McClusky, S. C., Potter, E. K., Pfeffer, J., McGirr, R., Purcell, A. P., Herring, T. A., and Montillet, J. P.: ANU GRACE Data Analysis: Orbit Modeling, Regularization and Inter-satellite Range Acceleration Observations, Journal of Geophysical Research-Solid Earth, 127, https://doi.org/10.1029/2021jb022489, 2022.

Gonzalez, A.: Measurement of Areas on a Sphere Using Fibonacci and Latitude-Longitude Lattices, Mathematical Geosciences, 42, 49-64, https://doi.org/10.1007/s11004-009-9257-x, 2010.

Ran, J., Ditmar, P., Liu, L., Xiao, Y., Klees, R., and Tang, X.: Analysis and Mitigation of Biases in Greenland Ice Sheet Mass Balance Trend Estimates From GRACE Mascon Products, Journal of Geophysical Research-Solid Earth, 126, https://doi.org/10.1029/2020jb020880, 2021.

Tregoning, P., McGirr, R., Pfeffer, J., Purcell, A., McQueen, H., Allgeyer, S., and McClusky, S. C.: ANU GRACE Data Analysis: Characteristics and Benefits of Using Irregularly Shaped Mascons, Journal of Geophysical Research-Solid Earth, 127, https://doi.org/10.1029/2021jb022412, 2022.

Fig. 1, the polar regions are distorted. Add plots using polar projection to better show these regions.

**Response:**

We sincerely appreciate your valuable comments. Following your suggestion, we have included polar projection maps illustrating the Mascon partitioning in both the Greenland and Antarctic regions (Figure 1 in the revised manuscript).

[Figure]

*Figure 1. Mascon partitioning of GCL-Mascon2024 solution*

Equ(4) \epsilon_0 is not explained.

**Response:**

We sincerely appreciate your valuable comments. We have supplemented the explanation of \epsilon_0, please kindly refer to the following text (Lines 227-229 of the revised manuscript).

*In this study, noise whitening filters $\boldsymbol{W}$, constructed based on postfit residuals derived from orbit and range-rate measurements using the autoregressive (AR) noise model implemented in the ARMASA toolbox (Broersen and Wensink, 1998; Broersen, 2000), are applied to transform frequency noise $\boldsymbol{\varepsilon}$ into Gaussian white noise $\boldsymbol{\varepsilon}_0$.*

Reference

Broersen, P. M. T. and Wensink, H. E.: Autoregressive model order selection by a finite sample estimator for the Kullback-Leibler discrepancy, Ieee Transactions on Signal Processing, 46, 2058-2061, https://doi.org/10.1109/78.700984, 1998.

Broersen, P. M. T.: Facts and fiction in spectral analysis, Ieee Transactions on Instrumentation and Measurement, 49, 766-772, https://doi.org/10.1109/19.863921, 2000.

Equ (5), there is no regularization factor in front of C_M?

**Response:**

We sincerely appreciate your valuable comments. We have modified this typo, please kindly refer to the following text (Lines 239-244 of the revised manuscript).

$$\hat{\boldsymbol{x}} = \left( \boldsymbol{A}^T \boldsymbol{P} \boldsymbol{A} + \mu \boldsymbol{C}_M \right)^{-1} \cdot \boldsymbol{A}^T \boldsymbol{P} \boldsymbol{L}, \tag{6}$$

*where $\hat{\boldsymbol{x}}$ represents the mascons to be estimated; $\boldsymbol{A}$ is the design matrix of partial derivatives; $\boldsymbol{L}$ is the residual vector which is obtained by subtracting the kinematic orbit or KBR measurements from the reference orbit positions or KBR data; $\boldsymbol{P}$ is the weight matrix derived from the inverse of the variance-covariance matrix $\boldsymbol{\Sigma}$ (refer to section 2.4); $\mu$ is the regularization factor; $\boldsymbol{C}_M$ is a diagonal constraint (or regularization) matrix of size $n \times n$, named the Mass Variation Regularization Constraint Normalized (MVRCN) Matrix; $n$ is the number of the mascons to be estimated.*

Fig. 3, I feel this map will be upscaled to the mascon-size resolution. If so, please also provide this map.

**Response:**

We sincerely appreciate your valuable comments. Following your suggestion, the regularization constraint matrix was upscaled to match the mascon scale. A comparative visualization between the original-scale matrix and its mascon-scaled counterpart is presented in Figure R12. As your comments pointed out, only the mascon-scaled regularization matrix (Figure R12-b) actively participated in the mascon solution recovery. Figure 3 has been updated to show the Mass Variation Regularization Constraint Normalized (MVRCN) matrix at mascon resolution in the revised manuscript.

[Figure]

Figure R12. The Mass Variation Regularization Constraint Normalized (MVRCN) Matrix used in the GCL-Mascon2024 recovery framework: (a) original resolution vs. (b) Mascon-scaled resolution

*It should be explained how the values in oceans are derived.*

***Response:***

*We sincerely appreciate your valuable comments. Following your suggestion, we have*

*explained how the values in oceans are derived in the revised manuscript (Lines 158-162 of the revised manuscript). Please kindly refer to the following text for more details.*

*Following a standardized processing workflow (Watkins et al., 2015; Save et al., 2016; Loomis et al., 2019; Tregoning et al., 2022), the uncorrected mascon solutions (i.e.,* $\text{MASCON}_{Uncorrected}$ *, we will return to that point in Sect. 2.5) are systematically integrated with the aforementioned corrected components to generate corrected mascon grids. The formula to generate the corrected mascon grid is*

$$\text{MASCON}_{Corrected} = \text{MASCON}_{Uncorrected} - \text{MASCON}_{C_{20}} + \text{SLR}_{C_{20}} + \text{DEG1} - \text{GIA} + \text{GAD} . \quad (4)$$

Reference

Loomis, B. D., Luthcke, S. B., and Sabaka, T. J.: Regularization and error characterization of GRACE mascons, Journal of Geodesy, 93, 1381-1398, https://doi.org/10.1007/s00190-019-01252-y, 2019.

Save, H., Bettadpur, S., and Tapley, B. D.: High-resolution CSR GRACE RL05 mascons, Journal of Geophysical Research-Solid Earth, 121, 7547-7569, https://doi.org/10.1002/2016jb013007, 2016.

Tregoning, P., McGirr, R., Pfeffer, J., Purcell, A., McQueen, H., Allgeyer, S., and McClusky, S. C.: ANU GRACE Data Analysis: Characteristics and Benefits of Using Irregularly Shaped Mascons, Journal of Geophysical Research-Solid Earth, 127, https://doi.org/10.1029/2021jb022412, 2022.

Watkins, M. M., Wiese, D. N., Yuan, D.-N., Boening, C., and Landerer, F. W.: Improved methods for observing Earth's time variable mass distribution with GRACE using spherical cap mascons, Journal of Geophysical Research-Solid Earth, 120, 2648-2671, https://doi.org/10.1002/2014jb011547, 2015.

L240, I don't get why there should be a constraint based on topography in ice sheets.

**Response:**

We sincerely appreciate your valuable comments. We use the Greenland Ice Sheet as an example to explain why the constraint matrix is designed based on the topography of the ice sheet. The GIS mass evolution is dominated by the signal from the coastal margins of the ice sheet (Luthcke et al., 2013). Below 2000 m elevation, significant coastal mass loss is primarily driven by ice discharge from fast-flowing, marine-terminating outlet glaciers in the northwest, southwest, and southeastern regions of the Greenland Ice Sheet (Moon et al., 2012); above this elevation threshold, no statistically

significant mass variations are observed (Luthcke et al., 2013). In the GSFC mascon solution (Luthcke et al., 2013; Loomis et al., 2019), the Greenland Ice Sheet is partitioned into two constraint regions based on elevation thresholds: (1) areas below 2000 m and (2) areas above 2000 m. Figure R13 depicts the Greenland Ice Sheet topography (Figure R13-a) and the mascon-scale regularization matrix (Figure R13-b) used in the GCL-Mascon2024 recovery framework. A notable spatial alignment is observed between regions with stronger regularization constraints (dark blue areas in Figure R13-b) and the 2000 m elevation contour zone of the ice sheet.

[Figure]

Figure R13. Comparison of Greenland Ice Sheet topography and regularization constraint matrix: (a) topographic map, (b) regularization constraint matrix

Reference

Loomis, B. D., Luthcke, S. B., and Sabaka, T. J.: Regularization and error characterization of GRACE mascons, Journal of Geodesy, 93, 1381-1398, https://doi.org/10.1007/s00190-019-01252-y, 2019.

Luthcke, S. B., Sabaka, T. J., Loomis, B. D., Arendt, A. A., McCarthy, J. J., and Camp, J.: Antarctica, Greenland and Gulf of Alaska land-ice evolution from an iterated GRACE global mascon solution, Journal of Glaciology, 59, 613-631, https://doi.org/10.3189/2013JoG12J147, 2013.

Moon, T., Joughin, I., Smith, B., and Howat, I.: 21st-Century Evolution of Greenland Outlet Glacier Velocities, Science, 336, 576-578, https://doi.org/10.1126/science.1219985, 2012.

L244, why in equ (5) does the L-curve take an effect? I suppose there is a regularization factor. Besides, is the regularization factor different month by month? If so, please show it. Third, please give some examples illustrating the sensitivity of results to different regularization factors.

**Response:**

We sincerely appreciate your valuable comments.

**Regarding your first comment**, "why in equ (5) does the L-curve take an effect? I suppose there is a regularization factor.", we have modified this typo, please kindly refer to the following text (Lines 239-244 of the revised manuscript).

$$\hat{x} = \left( A^T P A + \mu C_M \right)^{-1} \cdot A^T P L , \qquad (6)$$

*where $\hat{x}$ represents the mascons to be estimated; $A$ is the design matrix of partial derivatives; $L$ is the residual vector which is obtained by subtracting the kinematic orbit or KBR measurements from the reference orbit positions or KBR data; $P$ is the weight matrix derived from the inverse of the variance-covariance matrix $\Sigma$ (refer to section 2.4); $\mu$ is the regularization factor; $C_M$ is a diagonal constraint (or regularization) matrix of size $n \times n$, named the Mass Variation Regularization Constraint Normalized (MVRCN) Matrix; $n$ is the number of the mascons to be estimated.*

**Regarding your second comment**, "Besides, is the regularization factor different month by month? If so, please show it.", we have supplemented the explanation about the different regularization factors for each month. Please kindly refer to the following text (Lines 252-255 in the revised manuscript).

*We employ the L-curve method to determine the appropriate regularization factor $\mu$, employing monthly-varying factor values to ensure that the resulting regularization matrix is sufficiently tight to suppress noise yet loose enough to allow the mascons to adjust to their optimal values.*

**Regarding your third comment**, "Third, please give some examples illustrating the

sensitivity of results to different regularization factors.", we have supplemented the mascon solutions under three distinct regularization scenarios: under-regularized (insufficient constraint), appropriately regularized (balanced constraint), and over-regularized (excessive constraint). This comparative analysis has been incorporated as Figure R14 to demonstrate the sensitivity of our solutions to different regularization factors. We would like to clarify that the following figure shows the uncorrected mascon results, i.e., $\text{MASCON}_{Uncorrected}$ of Eq. (4).

[Figure]

Figure R14. Mascon results with three regularization scenarios:(a) under-regularized constraint, (b) appropriately regularized constraint, and (c) over-regularized constraint.

**Community Comment #1**

This paper presents some novel work for a new mascon result, particularly in the design of regularization matrix.

**Response:**

Thank you very much for your constructive comments on our manuscript. There is no doubt that these comments are valuable and very helpful for revising and improving our manuscript. Below is the point-by-point response to the specific remarks.

I woud like to know the underlying considerations behind different resolutions for ocean and land regions ( 400×400 km vs. 300×300 km).

**Response:**

We sincerely appreciate your valuable comments. Our design considerations for the dual-resolution strategy in ocean and land regions are as follows.

1. Land Mascon Resolution (300×300 km)

Over the land, the mascon size (300×300 km) aligns with GRACE's effective spatial resolution (~300 km), ensuring optimal recovery of surface mass transport signals (e.g., hydrology, ice sheet changes).

2. Ocean Mascon Resolution (400×400 km)

During the initial mascon determination, Atmospheric and Ocean De-aliasing models (i.e., AOD1B) are applied to mitigate high-frequency signals in background force modeling. AOD1B product provides a priori information about temporal variations in the Earth's gravity field caused by global mass variability in the atmosphere and ocean. However, there are still residual unmodeled high-frequency signals and errors over the ocean. These residuals are analogous to those in temporal gravity field spherical harmonic solutions (i.e., L1b -> L2), where open-ocean residual analysis is a standard approach to evaluate the accuracy of different spherical harmonic solutions (e.g., Darbeheshti et al., 2024; Zhou et al., 2024). To absorb such uncertainties and minimize their propagation into land signals, we intentionally defined ocean mascons with a coarser resolution (400×400 km). Furthermore, employing coarser-resolution oceanic mascons serves to minimize the parameter space and enhance the numerical stability of the inverse problem.

Reference

Darbeheshti, N., Lasser, M., Meyer, U., Arnold, D., and Jaggi, A.: AIUB-GRACE gravity field solutions for G3P: processing strategies and instrument parameterization, Earth System Science Data, 16, 1589-1599, https://doi.org/10.5194/essd-16-1589-2024, 2024.

Zhou, H., Zheng, L., Li, Y., Guo, X., Zhou, Z., and Luo, Z.: HUST-Grace2024: a new GRACE-only gravity field time series based on more than 20 years of satellite geodesy data and a hybrid processing chain, Earth System Science Data, 16, 3261-3281, https://doi.org/10.5194/essd-16-3261-2024, 2024.

The design of the MVRCN matrix lacks specific explanation for oceanic regions, and similarly, analysis of the results.

**Response:**

We sincerely appreciate your valuable comments. Following your suggestion, we have explained how the values in oceans are derived in the revised manuscript (Lines 158-162 of the revised manuscript) and the analysis of the ocean signals (Lines 510-517 of the revised manuscript).

*Following a standardized processing workflow (Watkins et al., 2015; Save et al., 2016; Loomis et al., 2019; Tregoning et al., 2022), the uncorrected mascon solutions (i.e., $\mathrm{MASCON}_{Uncorrected}$, we will return to that point in Sect. 2.5) are systematically integrated with the aforementioned corrected components to generate corrected mascon grids. The formula to generate the corrected mascon grid is*

$$\mathrm{MASCON}_{Corrected} = \mathrm{MASCON}_{Uncorrected} - \mathrm{MASCON}_{C_{20}} + \mathrm{SLR}_{C_{20}} + \mathrm{DEG1} - \mathrm{GIA} + \mathrm{GAD} . \quad (4)$$

*GRACE satellite gravity measurements over oceanic regions directly correspond to ocean bottom pressure variations at spatial scales of ~300 km (Watkins et al., 2015). Figure 13 illustrates the time series of basin mass variations derived from different mascon solutions. To assess the quality of our solutions for ocean signals, we compute the correlation coefficients between GCL-Mascon2024 and the RL06 mascon solutions released by GSFC, CSR, and JPL. The resulting correlations are 95.7%, 98.0%, and 98.2%, respectively, indicating a high level of consistency between our products and official mascon products.*

[Figure]

*Figure 13. Comparison of GRACE-derived mass anomaly time series (expressed in equivalent water height, EWH) over the global sea from different mascon solutions.*

Figure 8, Panel (b): y-axis label may be corrected from "mE/Hz1/2" to "m/Hz1/2"; Panel (d): to "m/s/Hz1/2".

**Response:**

We sincerely appreciate your valuable comments. We have corrected this typo in the revised version of the manuscript. Please kindly refer to the following figure.

[Figure]

*Figure 2. Time series and power spectrum densities (PSD) of postfit residuals from orbit and KBR range rate*